# A Dual-Perspective Approach to Evaluating Feature Attribution Methods

**Yawei Li**[*]                                                    *yawei.li@stat.uni-muenchen.de*
*LMU Munich*
*Munich Center for Machine Learning*

**Yang Zhang**[*]                                                  *yangzhang@u.nus.edu*
*National University of Singapore*

**Kenji Kawaguchi**                                                *kenji@comp.nus.edu.sg*
*National University of Singapore*

**Ashkan Khakzar**                                                 *ashkan.khakzar@eng.ox.ac.uk*
*University of Oxford*

**Bernd Bischl**                                                   *bernd.bischl@stat.uni-muenchen.de*
*LMU Munich*
*Munich Center for Machine Learning*

**Mina Rezaei**                                                    *mina.rezaei@stat.uni-muenchen.de*
*LMU Munich*
*Munich Center for Machine Learning*

**Reviewed on OpenReview:** *https://openreview.net/forum?id=znlTP5RLur*

## Abstract

Feature attribution methods attempt to explain neural network predictions by identifying relevant features. However, establishing a cohesive framework for assessing feature attribution remains a challenge. There are several views through which we can evaluate attributions. One principal lens is to observe the effect of perturbing attributed features on the model's behavior (i.e., faithfulness). While providing useful insights, existing faithfulness evaluations suffer from shortcomings that we reveal in this paper. To address the limitations of previous evaluations, in this work, we propose two new perspectives within the faithfulness paradigm that reveal intuitive properties: *soundness* and *completeness*. Soundness assesses the degree to which attributed features are truly predictive features, while completeness examines how well the resulting attribution reveals all the predictive features. The two perspectives are based on a firm mathematical foundation and provide quantitative metrics that are computable through efficient algorithms. We apply these metrics to mainstream attribution methods, offering a novel lens through which to analyze and compare feature attribution methods. Our code is provided at `https://github.com/sandylaker/soco.git`.

## 1 Introduction

Understanding predictions of machine learning models is a crucial aspect of trustworthy machine learning across diverse fields, including medical diagnosis (Bernhardt et al., 2022; Khakzar et al., 2021c;b), drug discovery (Callaway, 2022; Jiménez-Luna et al., 2020; Gündüz et al., 2023; 2024), and autonomous driving (Kaya et al., 2022; Can et al., 2022). Feature attribution, indicating the contribution of each feature to

---

[*]Equal contribution.

a model prediction, serves as a fundamental approach to interpreting neural networks. However, the outcomes from various feature attribution methods can be inconsistent for a given input (Krishna et al., 2022), necessitating distinct evaluation metrics to gauge how well a feature attribution elucidates the prediction.

Research has introduced various lenses through which we can evaluate attributions. One lens is assessing the methods through sanity checks (Adebayo et al., 2018). For example, by checking if the attribution changes if network parameters are randomized. Another evaluates attributions against ground truth features (Zhang et al., 2018; Yang et al., 2022; Zhang et al., 2023). Each lens reveals different insights. However, one lens is of particular interest in our study, evaluation via faithfulness. Faithfulness measures the degree to which the attributions mirror the relationships between features and the model's behavior. For instance, how changing (e.g., removing) attributed features affects the model's performance. This analysis has taken several forms, for instance by perturbing features according to their rankings and checking the immediate effect on output (Samek et al., 2016; Ancona et al., 2018), or first perturbing features, then re-training the network from scratch on the perturbed features (Hooker et al., 2019; Zhou et al., 2022; Rong et al., 2022). However, one common property exists between all these forms of faithfulness analysis. The evaluations solely consider the ranking of attribution values and disregard the attribution values.

In this work, we leverage the notion of considering the value of attributions in addition to the order and introduce two complementary perspectives within faithfulness: *soundness* and *completeness*. They serve as an evaluation of the alignment between attribution and predictive features. *Soundness* assesses the degree to which attributed features are truly predictive features, while *completeness* examines how well the resulting attribution map reveals all the predictive features. These proposed metrics work in tandem and reflect different aspects of feature attribution methods. We first motivate the work by revealing issues within existing faithfulness evaluations. We further see that by considering the attribution value in addition to the order, our metrics are more distinctive compared to existing methods, enabling a more precise differentiation between attribution methods. Through extensive validation and benchmarking, we verify the correctness of the proposed metrics and showcase our metrics' potential to shed light on existing attribution methods.

## 2 Related work

### 2.1 Feature attribution methods

Attribution methods explain a model by assigning a score to each input feature, indicating the importance of that feature to the model's prediction. These methods can be categorized as follows:

- Gradient-based methods: These methods (Simonyan et al., 2014; Baehrens et al., 2010; Springenberg et al., 2015; Khakzar et al., 2021a; Zhang et al., 2018; Shrikumar et al., 2017) generate attributions based on variants of back-propagation rules. For instance, Simonyan et al. (2014) employs the absolute values of gradients to determine feature importance, whereas DeepLIFT (Shrikumar et al., 2017) calculates attributions by decomposing the output prediction of a neural network into contributions from each input feature. It uses a reference activation to compare against actual activations, measuring the difference between the neuron's activation for a given input and its activation at the reference point. This difference is then propagated backward through the network, layer by layer, using a set of rules specific to the activation functions and network architecture.

- Hidden activation-based methods: CAM (Zhou et al., 2016) produces attribution maps for CNNs, which are equipped with global average pooling and a linear classification head. CAM multiplies the last CNN layer activations with the weights in the linear head associated with a target class, and then computing the weighted average. GradCAM (Selvaraju et al., 2017) further generalizes this approach by weighting the activation maps using the gradients, and then computing the weighted sum over the activation maps. Importantly, GradCAM does not require the network to have a global average pooling followed by a linear classification layer, thus can be applied to a broader range of neural network architectures.

- Shapley value-based methods: This class of methods approximates Shapley values (Shapley et al., 1953), considering features as cooperative players with different contributions to the predic-

tions. DeepSHAP, which combines ideas from DeepLIFT and SHAP (SHapley Additive exPlanations) (Lundberg & Lee, 2017), leverages SHAP's approach of using Shapley values. Specifically, DeepSHAP approximates these Shapley values by integrating DeepLIFT's rules for backpropagating contributions from the output to the inputs, while considering multiple reference values to account for the distribution of the input data, which enhances the accuracy and stability of the attributions. Integrated Gradients (IG) (Sundararajan et al., 2017) integrates the gradients of the network's output with respect to its inputs along a straight path from a baseline (typically a zero vector) to the actual input. This path integration helps in capturing the importance of each input feature across different scales. While not directly employing Shapley values, the integral in this method can be seen as an approximation of Shapley value calculations, attributing the output prediction fairly among all input features by considering their marginal contributions along the path. SmoothGrad (Smilkov et al., 2017) is an approach that reduces noise in IG by averaging the gradients of the input with small amounts of random noise added multiple times, enhancing the visual sharpness and interpretability of the attributions. SmoothGrad$^2$ extends this by squaring the gradients before averaging, which emphasizes larger gradient values more and can highlight features more strongly. VarGrad (Adebayo et al., 2018), another variant, focuses on the variance of the gradients with noise added, aiming to capture areas of the input space where the model's predictions are most sensitive to perturbations, thereby identifying potentially important features.

- Perturbation-based methods: This type of methods works by perturbing parts of the input or hidden activations and observing the impact on the model's prediction, specifically targeting those regions whose alternation leads to the most extreme change in output. Extremal Perturbation (Fong et al., 2019) employs optimization techniques to systematically search for a smooth mask identifying the most influential regions of an input image. IBA (Schulz et al., 2020) searches the smooth mask on hidden activations using information bottleneck. The core idea is viewing the masked hidden features as a random variable $Z$, and maximizing the mutual information $I(Y; Z)$ between target label $Y$ and features $Z$, while minimizing the mutual information $I(X; Z)$ between the feature $Z$ and input $X$. The optimized information bottleneck, parametrized by the optimal mask, only admits a subset of features through the downstream layers. These admitted features can best maintain the model's performance, therefore they are the most important features. InputIBA (Zhang et al., 2021) reveals hidden features typically have small spatial size (e.g., $7 \times 7$), hence the mask returned by IBA can suffer from over-blurriness when being resized to the input spatial size. Furthermore, putting the IBA at early layers suffers from over-estimate of the mutual information, as the features at early layers do not necessarily follow Gaussian distribution, violating the assumption employed by IBA. InputIBA solves this challenge by using Generative Adversarial Networks (Goodfellow et al., 2020; Arjovsky et al., 2017) to approximate the unknown hidden activation distribution, thereby producing more sharper and more detailed attribution maps.

## 2.2 Evaluation metrics for feature attribution methods

Existing metrics for assessing the faithfulness of attribution methods can be categorized as follows:

- Expert-grounded metrics: These metrics rely on human expertise to interpret and assess the quality of attribution maps produced by attribution methods. Experts visually inspect (Yang et al., 2022) these maps to determine if they highlight the relevant regions, as expected by the model's focus. For example, if a classifier identifies an image as containing a basketball, the attribution map should highlight the basketball itself. Another approach involves the "pointing game" (Zhang et al., 2018), where the goal is to see if the attribution method assigns the highest value to the pixel within the bounding box that localizes the object in question. Additionally, human-AI collaborative tasks (Nguyen et al., 2021) gauge the effectiveness of attribution maps in aiding human classification efforts. While these methods incorporate human judgment, their outcomes can be subjective and may lack consistency.

- Functional-grounded metrics: These metrics, which include works by (Petsiuk et al., 2018; Samek et al., 2016; Ancona et al., 2018; Hooker et al., 2019; Rong et al., 2022), operate by removing input

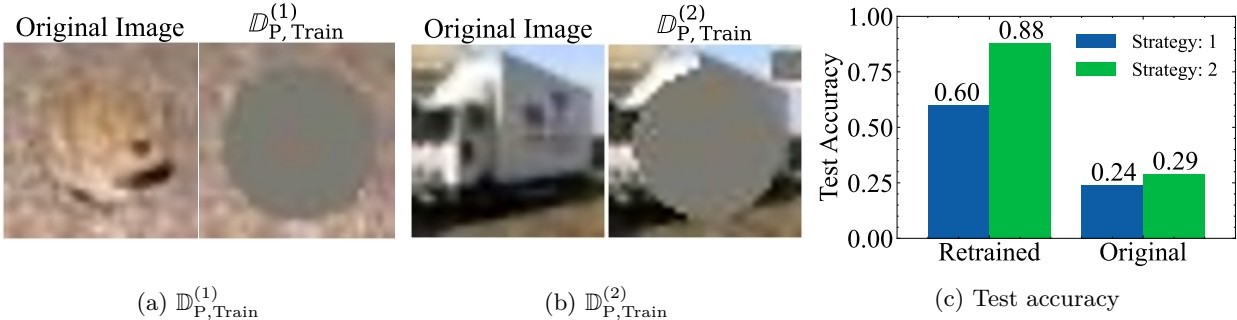

(a) $\mathbb{D}^{(1)}_{\mathrm{P,Train}}$  (b) $\mathbb{D}^{(2)}_{\mathrm{P,Train}}$  (c) Test accuracy

Figure 1: **Analysis of retraining-based metrics.** Compared to (a) $\mathbb{D}^{(1)}_{\mathrm{P,Train}}$, (b) $\mathbb{D}^{(2)}_{\mathrm{P,Train}}$ introduces an additional class-related spurious correlation during perturbation, visible in the upper-right region of the sample. (c) Despite equivalent removal of informative features (central portions of images) using both perturbation strategies, the two retrained models demonstrate different test accuracy (0.66 vs. 0.88), suggesting that the test accuracy of the *retrained* model does not accurately reflect the quantity of information removal.

features and measuring changes in the model's predictions. The underlying premise is that removing an important feature should result in a significant prediction change. Initially introduced by Insertion/Deletion (Samek et al., 2016), two feature removal orders are considered: Most Relevant First (MoRF), which starts with the feature having the highest attribution value, and Least Relevant First (LeRF), which begins with the feature having the least attribution value. Plotting the probability of the target class against the number of removed features results in two distinct curves for MoRF and LeRF. A faithful attribution method should correctly rank feature importance, resulting in a steep initial drop in the MoRF curve, followed by a plateau, whereas the LeRF curve should show a plateau at the start and a sharp drop towards the end. Hooker et al. (2019) introduced Remove and Retrain (ROAR) to mitigate potential adversarial effects of feature removal by retraining the model on perturbed datasets. ROAD (Rong et al., 2022) highlighted that perturbation masks could inadvertently leak class information, potentially misrepresenting feature importance. More recently, Zhou et al. (2022) proposed a method to inject "ground truth" features into the training dataset, forcing the model to learn exclusively from these features and then testing the ability of attribution methods to recognize them.

- Others: Khakzar et al. (2022) introduced empirical evaluation of axioms, and Adebayo et al. (2018) introduced sanity checks for saliency maps, further diversifying the landscape of attribution method evaluation.

## 3 Analysis of prior evaluation metrics

In this section, an empirical analysis of various feature attribution evaluations is conducted, with the objective of delineating the advantages and disadvantages of existing evaluation metrics. Later, we design our metrics with care to circumvent the potential pitfalls in the previous evaluations.

### 3.1 Retraining-based evaluation metrics

Retraining-based evaluations, such as ROAR (Hooker et al., 2019), involve retraining the model on a perturbed dataset and measuring the accuracy of the retrained model on a perturbed test set. A sharper decrease in accuracy suggests a greater information loss resulting from the perturbation of attributed features, thus indicating better feature attribution. Since the model is retrained, it is not subject to OOD effects instigated by perturbation, as observed in Insertion/Deletion. However, the perturbation may give rise to other spurious features when the original ones are removed. The model might learn these newly introduced features, as there are no stringent constraints in the learning process to prevent the model from doing so. Therefore, if the retrained model leverages these spurious features rather than relying exclusively

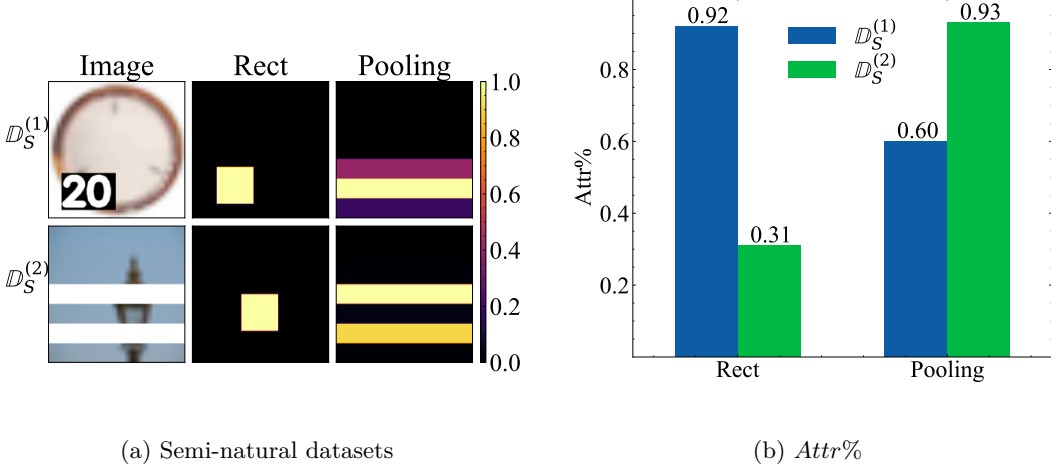

(a) Semi-natural datasets

(b) *Attr%*

Figure 2: **Analysis of evaluation on semi-natural datasets.** (a) Designed semi-natural datasets and attribution maps from crafted attribution methods. (b) Each method excels on the dataset for which it has prior knowledge, but it underperforms on the other.

on the remaining ones to make predictions, the testing accuracy might not accurately represent the extent of information loss.

We illustrate the issue of retraining via a falsification experiment, following the ROAR setting. We perturb 70% of pixels in each image in the CIFAR-10 (Krizhevsky et al., 2009a) training and test datasets. The model is then retrained on the perturbed training set and evaluated on the perturbed test set. Specifically, we employ two perturbation strategies: Strategy 1, which perturbs a central circular region to remove class-related objects (Figure 1a), and Strategy 2, which perturbs the image center and a small edge region based on the class label (Figure 1b), introducing a spurious class correlation. We refer to the **P**erturbed **Train**ing set using strategy **1** as $\mathbb{D}^{(1)}_{\mathrm{P,Train}}$. Analogously, we have $\mathbb{D}^{(2)}_{\mathrm{P,Train}}$, $\mathbb{D}^{(1)}_{\mathrm{P,Test}}$, and $\mathbb{D}^{(2)}_{\mathrm{P,Test}}$. We then train a model on the original dataset, retrain it on $\mathbb{D}^{(1)}_{\mathrm{P,Train}}$ and $\mathbb{D}^{(2)}_{\mathrm{P,Train}}$, and report the accuracy of both the original and retrained models on $\mathbb{D}^{(1)}_{\mathrm{P,Test}}$ and $\mathbb{D}^{(2)}_{\mathrm{P,Test}}$. Additional details are in Appendix C. As shown in Figure 1c, the original model performs poorly on $\mathbb{D}^{(1)}_{\mathrm{P,Test}}$ and $\mathbb{D}^{(2)}_{\mathrm{P,Test}}$ due to the perturbation on substantial class-related pixels. However, the two retrained models achieve distinct accuracy on their test sets. While the model retrained on $\mathbb{D}^{(1)}_{\mathrm{P,Train}}$ still performs badly on $\mathbb{D}^{(1)}_{\mathrm{P,Test}}$, the model retrained on $\mathbb{D}^{(2)}_{\mathrm{P,Train}}$ has almost 90% test accuracy on $\mathbb{D}^{(2)}_{\mathrm{P,Test}}$, as the latter model learns the spurious correlation introduced by perturbation. Despite both perturbation strategies notably disrupting the central informative part of an image, the model retrained on $\mathbb{D}^{(2)}_{\mathrm{P,Train}}$ still achieves high test accuracy. Therefore, spurious features can have a great impact on the evaluation outcome.

### 3.2 Evaluation on semi-natural datasets

If we have access to the features that are truly relevant to labels, we can compare them with attribution maps to evaluate attribution methods. Zhou et al. (2022) proposed training the model on a dataset with injected ground truth features. However, our subsequent experiment reveals that the evaluation outcome can be affected by the design of ground truth features. Moreover, results from semi-natural datasets may diverge from those on real-world datasets, as utilizing semi-natural datasets changes the original learning task.

The way we construct semi-natural datasets significantly influences the properties of the introduced ground truth, such as its size and shape. With prior knowledge of semi-natural dataset construction, we can tailor attribution methods to outperform others on this dataset. To illustrate this, we create two **S**emi-natural **D**ataset $\mathbb{D}^{(1)}_{\mathrm{S}}$ and $\mathbb{D}^{(2)}_{\mathrm{S}}$ from CIFAR-100 (Krizhevsky et al., 2009b). In the case of $\mathbb{D}^{(1)}_{\mathrm{S}}$, numeric watermarks

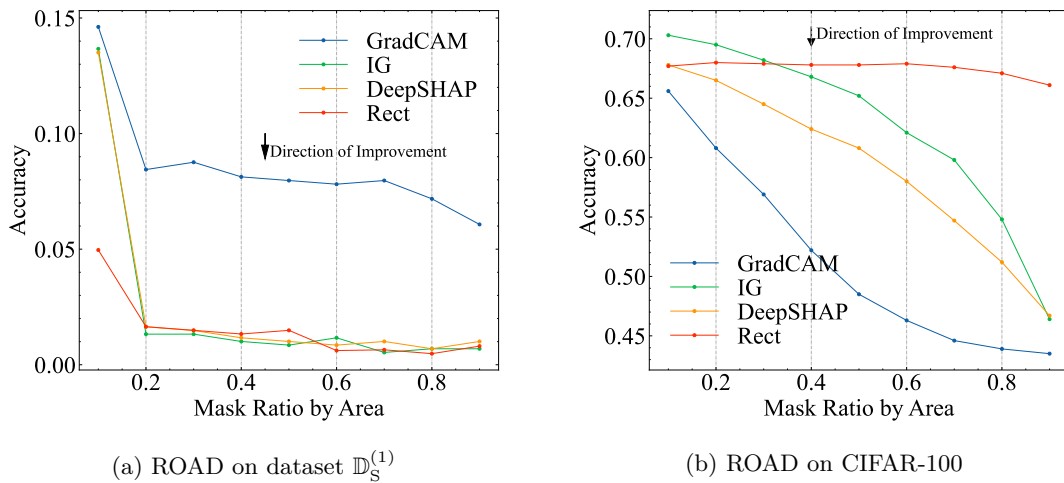

(a) ROAD on dataset $\mathbb{D}_S^{(1)}$                    (b) ROAD on CIFAR-100

Figure 3: **Evaluation on semi-natural datasets vs. on real-world datasets.** Evaluation results on a semi-natural and real dataset can be markedly different. On the semi-natural dataset $\mathbb{D}_S^{(1)}$, A "dummy" method *Rect* simply using the prior information about the dataset $\mathbb{D}_S^{(1)}$ performs the best, while it has the worst performance on CIFAR-100.

that correspond to class labels are injected (Figure 2a first row, left column), whereas for $\mathbb{D}_S^{(2)}$, each image is divided into seven regions, and stripes are inserted, acting as binary encoding for class labels (Figure 2a second row, left column). For example, watermarks are put in the 4th and 6th regions for class 40 (i.e., $0101000_2$). Additionally, we design *Rect* attribution method to take advantage of the prior knowledge that the watermarks in $\mathbb{D}_S^{(1)}$ are square patches, and we design *Pooling* attribution for $\mathbb{D}_S^{(2)}$ by exploiting the fact that a stripe watermark in $\mathbb{D}_S^{(2)}$ forms a rectangle spanning an entire row. The name "Pooling" comes from the operation of averaging attribution values within each row-spanning region in $\mathbb{D}_S^{(2)}$, then broadcasting this pooled value across the region. Further details can be found in Appendix D.2. Figure 2a visualizes attribution maps for *Rect* and *Pooling*. Using the *Attr%* metric (Zhou et al., 2022), we evaluate both methods on both datasets (Figure 2b). Each method excels on the dataset it was designed for but performs poorly on the other, demonstrating the inconsistency of evaluations on semi-natural datasets with different ground truth features.

Besides the insights from the previous experiment, we further show the inconsistency between evaluation results on semi-natural and real datasets. Due to the absence of ground truth on real datasets, we replace the *Attr%* metric with ROAD (Rong et al., 2022) to assess attribution methods. We utilize CIFAR-100 as the real dataset and evaluate four distinct attribution methods (details in Appendix D.1). The ROAD results on semi-natural dataset $\mathbb{D}_S^{(1)}$ and CIFAR-100 are depicted in Figure 3a and Figure 3b, respectively. These figures demonstrate that our tailored *Rect* method excels in ROAD evaluation on the semi-natural dataset, particularly at high mask ratios, but underperforms on CIFAR-100, indicating the bias in evaluations on semi-natural datasets. Similarly, non-customized attribution methods like GradCAM, IG, and DeepSHAP exhibit inconsistent performance across the two datasets, underscoring that evaluation on semi-natural and real datasets can yield distinct results.

## 3.3 Order-based evaluation metrics

Many evaluation metrics for feature attribution methods, such as those described by ROAD (Rong et al., 2022) and Insertion/Deletion (Petsiuk et al., 2018), operate by incrementally perturbing features based on their sorted indices, which are derived from the feature attribution values. These perturbed inputs are then fed into the model to compute predictions, which are compared with those from the original input. The variation in predictions serves as an indicator of the importance of the perturbed features. Despite the popularity of these order-based metrics, we emphasize that they only assess the relative attribution order of

the features——that is, whether one feature is more important than another. They do not account for the magnitude of differences between the attribution values of the features. Consider a synthetic example where an input has three features, $[x_1, x_2, x_3]^\top$, and attribution method A yields an attribution map $[1.0, 2.0, 3.0]^\top$, while method B produces $[1.0, 1.1, 100]^\top$. Employing an order-based metric like Deletion (Petsiuk et al., 2018), the removal order for both maps would be identical, i.e., $x_3$, $x_2$, $x_1$. This would result in identical output variations for each removal step and produce two identical curves when plotting output change against the removed feature index, ultimately leading to the same AUROC value for both methods. However, the semantic information conveyed by the attribution maps differs significantly: method A suggests that $x_3$ is slightly more important than $x_1$ and $x_2$, whereas method B indicates that $x_3$ is far more important. In summary, although order-based metrics can compare the relative importance of features, their ability to distinguish attribution values is limited.

## 4 Method

As highlighted in Section 3, both model retraining and the construction of semi-natural datasets present shortcomings that hurting the faithfulness of evaluation. In response, we have developed an alternative approach that *avoids model retraining and the creation of additional datasets*. Our method employs the fixed trained model and does not inject "ground truth" features into the dataset. Consequently, this approach is free from the shortcomings identified in Section 3. Specifically, our approach originates from the observation that misalignment between attributed features and ground truth predictive features occurs in two distinct ways: (1) non-predictive features are incorrectly attributed; (2) predictive features receive zero attribution. Motivated by this observation, we formalize two essential properties of feature attribution: *attribution soundness* and *attribution completeness*. The combined evaluation of these two properties offers a more refined and comprehensive assessment of the faithfulness of an attribution method.

### 4.1 Problem formulation

Our evaluation scenario is restricted to a specific model and dataset. This focus stems from our demonstration in Section 3 that the performance of attribution methods can significantly vary across different models and datasets. To aid readers, we include a table of mathematical notations in this work in Appendix A. Given a model $f$ that takes a set of features $\mathcal{F}$ as input, we define the predictive information measurement function $\varphi$ and attribution method $\eta$ as follows:

**Definition 4.1** (Predictive information measurement $\varphi$). For a feature set $\mathcal{F}$ and a feature $F \in \mathcal{F}$, $\varphi(F, \mathcal{F}; f) \in \mathbb{R}_{\geq 0}$ represents the amount of predictive information of $F$.

**Definition 4.2** (Attribution method $\eta$). An attribution method $\eta$ for a model $f$ is a function that assigns a value $\eta(F, \mathcal{F}; f) \in \mathbb{R}_{\geq 0}$ as the attribution to each feature $F$ in the feature set $\mathcal{F}$. This value quantifies the importance or contribution of feature $F$ to the predictions made by the model $f$.

Definition 4.1 and Definition 4.2 establish the frameworks for measuring true information and the attribution process within a model, using functions. We provide a concrete illustrative example for better understanding. Consider model $f$ as an image classification model that takes an image as input. Here, the feature set $\mathcal{F}$ represents the input image, and the feature $F$ is a pixel in the input image. Since we constrain our discussion to a particular model and input data, we omit parameters $f$ and $\mathcal{F}$ and use $\eta(F)$ or $\varphi(F)$ in the following text for notation simplicity. For a model $f$, there only exists a unique $\varphi$ that measures the predictive information for the model. However, $\varphi$ is inaccessible since knowing it requires a complete understanding of the model and its inner working mechanism. In contrast, there exist numerous possible attribution methods $\eta$. We can define the optimality of an attribution method as functional equivalence:

**Definition 4.3** (Optimality of attribution method). An attribution method $\eta$ is optimal, if $\eta$ equals $\varphi$.

However, it is challenging to directly compare the attribution method $\eta$ with the predictive information measurement $\varphi$ because their analytical forms are usually not accessible. Instead, we assess their outcomes. To do so, we focus on two specific subsets of the feature set $\mathcal{F}$. For a model $f$, a feature set $\mathcal{F}$, and an attribution method $\varphi$, the predictive feature set $\mathcal{I}$ and the attributed feature set $\mathcal{A}$ are defined as follows:

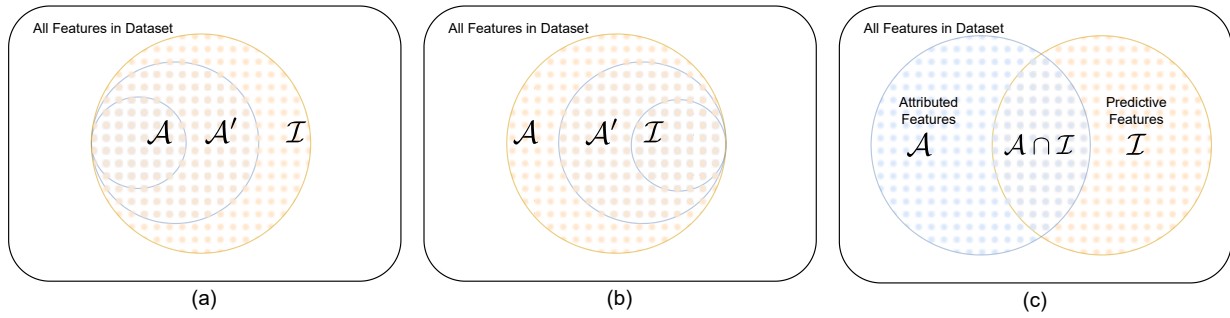

Figure 4: **Graphical demonstration** for a better understanding of (a) the relationship between two attributions ($\mathcal{A}$ and $\mathcal{A}'$). Although $\mathcal{A}$ and $\mathcal{A}'$ have equal soundness (1.0 in this case), $\mathcal{A}'$ has higher completeness. (b) Although $\mathcal{A}$ and $\mathcal{A}'$ have equal completeness, $\mathcal{A}'$ has higher soundness. (c) We compare $\mathcal{A} \cap \mathcal{I}$ with $\mathcal{A}$ and $\mathcal{I}$ to measure soundness and completeness.

**Definition 4.4** (Predictive feature set $\mathcal{I}$)**.** $\mathcal{I} \subseteq \mathcal{F}$ is a predictive feature set if $\mathcal{I} = \{F \in \mathcal{F} \mid \varphi(F) > 0\}$.

**Definition 4.5** (Attributed feature set $\mathcal{A}$)**.** $\mathcal{A} \subseteq \mathcal{F}$ is an attributed feature set if $\mathcal{A} = \{F \in \mathcal{F} \mid \eta(F) > 0\}$.

In essence, $\mathcal{I} \subseteq \mathcal{F}$ represents the features that are utilized by the model for decision-making, while $\mathcal{A} \subseteq \mathcal{F}$ encompasses features identified as significant by the attribution method $\eta$. Therefore, we can compare the alignment between $\mathcal{A}$ and $\mathcal{I}$ to determine the faithfulness of the attribution method $\eta$ on a certain feature set.

**Definition 4.6** (Optimality of attributed feature set $\mathcal{A}$)**.** Given a predictive feature set $\mathcal{I}$, an attributed feature set $\mathcal{A}$ is sound if $\mathcal{A} \subseteq \mathcal{I}$, complete if $\mathcal{I} \subseteq \mathcal{A}$, and optimal (sound and complete) if $\mathcal{A} = \mathcal{I}$.

Definition 4.6 outlines the necessary conditions for an attributed feature set to be deemed optimal, namely, the set must be both sound and complete. This condition is unique in that the elements in $\mathcal{A}$ and $\mathcal{I}$—or, equivalently, the feature indices—must match exactly. However, simply comparing the feature indices of two sets is insufficient to fully assess the faithfulness of attribution. This is because different attribution methods may assign varying values to the same feature, and a feature may carry different predictive information across various trained models. Therefore, it is crucial to consider both the attribution value and the predictive information associated with a feature to further assess the alignment between $\mathcal{A}$ and $\mathcal{I}$. To this end, we first introduce the operator $| \cdot |_g$ to measure the "cardinality" of a set. Subsequently, we define two metrics to gauge the soundness and completeness of $\mathcal{A}$.

**Definition 4.7** (Operator $| \cdot |_g$)**.** Given a feature set $\mathcal{F}$ and a function $g$, $|\mathcal{F}|_g = \sum_{F \in \mathcal{F}} g(F)$. For $\emptyset$, we define $|\emptyset|_g = 0$.

In other words, for a set of features $\mathcal{F}$, $|\mathcal{F}|_\eta$ computes the total attribution of all features in $\mathcal{F}$ as determined by the attribution method $\eta$, while $|\mathcal{F}|_\varphi$ computes the total amount of predictive information in $\mathcal{F}$. Following our earlier definitions, we can finally define the two properties of attribution:

**Definition 4.8** (Soundness)**.** For a set of attributed features $\mathcal{A}$, the soundness is the ratio $\frac{|\mathcal{A} \cap \mathcal{I}|_\eta}{|\mathcal{A}|_\eta}$.

**Definition 4.9** (Completeness)**.** For a set of attributed features $\mathcal{A}$, the completeness is the ratio $\frac{|\mathcal{A} \cap \mathcal{I}|_\varphi}{|\mathcal{I}|_\varphi}$.

Note that we use two operators separately in Definition 4.8 and Definition 4.9. Soundness measures how much of the attributed features actually contain predictive information. Complementary to soundness, completeness evaluates how comprehensively the attribution captures all predictive features. Both metrics are necessary to evaluate an attribution method. Figure 4a illustrates two sets of attributed features with equivalent soundness but divergent levels of completeness, whereas Figure 4b depicts two sets of attributed features that share equal completeness but exhibit different soundness. Figure 4c illustrates the relationship between $\mathcal{A} \cap \mathcal{I}$, $\mathcal{A}$, and $\mathcal{I}$.

Given our formulation, one challenge of measuring soundness and completeness is that we have no information about predictive features $\mathcal{I}$. Therefore, it is infeasible to directly calculate either $|\mathcal{A} \cap \mathcal{I}|_\eta$ or $|\mathcal{A} \cap \mathcal{I}|_\varphi$. Despite

---

**Algorithm 1** Soundness evaluation at predictive level $v$

---

1: **Input:** labeled dataset $\mathbb{D} = \{x^{(i)}, y^{(i)}\}_{i=1}^N$ with attribution maps $\{\mathcal{A}^{(i)}\}_{i=1}^N$, model $f$, predictive level $v$, accuracy threshold $\epsilon$.

2: **Initialize** $s = 0$, $\{\hat{\mathcal{A}}^{(i)}\}_{i=1}^N = \{\mathcal{A}_{\text{inc}}^{(i)}\}_{i=1}^N = \emptyset$;          `// Start with empty` $\mathcal{A}_{\text{inc}}$.

3: **while** $s < v$ **do**

4:      $\{\mathcal{A}_{\text{inc}}^{(i)}\}_{i=1}^N$, $\{\Delta\mathcal{A}^{(i)}\}_{i=1}^N = \text{Expand}(\{\mathcal{A}_{\text{inc}}^{(i)}\}_{i=1}^N)$;      `// Expand by adding the features with the`
                                                                                `highest attribution from` $\mathcal{A} \setminus \mathcal{A}_{\text{inc}}$,
                                                                          `and record newly included features.`

5:      $\tilde{s} = \text{Accuracy}(f, \{\mathcal{A}_{\text{inc}}^{(i)}, y^{(i)}\}_{i=1}^N)$;

6:      **if** $\tilde{s} - s < \epsilon$ **then**

7:          $\{\hat{\mathcal{A}}^{(i)}\}_{i=1}^N = \{\hat{\mathcal{A}}^{(i)} \cup \Delta\mathcal{A}^{(i)}\}_{i=1}^N$;

8:      **end if**

9:      $s = \tilde{s}$;

10: **end while**

11: $\{\mathcal{A}^{*(i)}\}_{i=1}^N = \{\mathcal{A}_{\text{inc}}^{(i)} \setminus \hat{\mathcal{A}}^{(i)}\}_{i=1}^N$;      `// Find` $\mathcal{A}^*$ `by excluding accumulated` $\mathcal{S} \cap (\mathcal{F} \setminus \mathcal{I})$.

12: $\{q^{(i)}\}_{i=1}^N = \{|\mathcal{A}^{*(i)}|_\eta / |\mathcal{A}_{\text{inc}}^{(i)}|_\eta\}_{i=1}^N$;

13: **Return** $\bar{q} = \frac{1}{N} \sum_{i=1}^N q^{(i)}$      `// Average soundness for all samples.`

---

this absence of explicit knowledge about the model's predictive features, we can leverage the model to provide indirect indications. We make the following assumptions throughout the work:

**Assumption 4.10.** Given a dataset $\mathbb{D}$, let $\mathcal{F}$ be the set of all input features in $\mathbb{D}$, $f$ be a model, and $\rho$ be a performance metric to assess the performance of model $f$.

$$\forall \mathcal{F}_1, \mathcal{F}_2 \subseteq \mathcal{F}, \text{ if } \rho(f(\mathcal{F}_1)) < \rho(f(\mathcal{F}_2)), \text{ then } |\mathcal{I} \cap \mathcal{F}_1|_\varphi < |\mathcal{I} \cap \mathcal{F}_2|_\varphi,$$

where $\mathcal{I} \subseteq \mathcal{F}$ is the set of predictive features for the model $f$.

Based on Assumption 4.10, we can compare $|\mathcal{I} \cap \mathcal{F}_1|_\varphi$ with $|\mathcal{I} \cap \mathcal{F}_2|_\varphi$ using the model performance given two sets of features $\mathcal{F}_1$ and $\mathcal{F}_2$. Specifically, while model performance is useful for identifying feature sets that contain more information, it is not suitable for quantifying the precise difference in information. Such a measurement requires much stronger assumptions than Assumption 4.10. We conjecture Assumption 4.10 to be true for models converged in training. Next, we show how to measure soundness and completeness based on our definitions and Assumption 4.10.

## 4.2 Soundness evaluation

Owing to the intractability of $|\mathcal{A} \cap \mathcal{I}|_\eta$, the direct calculation of the soundness of $\mathcal{A}$ in a single step is infeasible. However, we shall demonstrate that an iterative approach can effectively gauge the soundness for a subset $\mathcal{A}_{\text{inc}} \subseteq \mathcal{A}$ that is **inc**luded within the input. To ensure a fair evaluation across various attribution methods, we define $\mathcal{A}_{\text{inc}}$ in our implementation as a subset of the features that possess the highest attribution values, satisfying the condition $\rho(f(\mathcal{A}_{\text{inc}})) = v > 0$. In essence, this subset encompasses predictive features that cumulatively attain a specified *predictive level* $v$. A predictive level can be measured in various ways. For instance, in a classification task, it can be measured by Accuracy. The subsequent theorem shows the methodology for identifying the truly predictive portion within $\mathcal{A}_{\text{inc}}$, as well as the means to compute the soundness.

**Theorem 4.11.** *Given a set* $\mathcal{A}_{\text{inc}} \subseteq \mathcal{A}$ *and* $\mathcal{A}_{\text{inc}} \cap \mathcal{I} \neq \emptyset$, *suppose that* $\mathcal{S}_v(\mathcal{A}_{\text{inc}}) = \{\mathcal{S} \subseteq \mathcal{A}_{\text{inc}} : \rho(f(\mathcal{S})) = \rho(f(\mathcal{A}_{\text{inc}}))\}$ *is not empty for any* $\mathcal{A}_{\text{inc}} \subseteq \mathcal{A}$. *Let* $\mathcal{A}^* \in \arg\min_{\mathcal{S} \in \mathcal{S}_v(\mathcal{A}_{\text{inc}})} |\mathcal{S}|_\eta$. *Then, the soundness of* $\mathcal{A}_{\text{inc}}$ *is* $\frac{|\mathcal{A}^*|_\eta}{|\mathcal{A}_{\text{inc}}|_\eta}$.

**Proof**: For any $\mathcal{S} \in \mathcal{S}_v(\mathcal{A}_{\text{inc}})$ including $\mathcal{A}^*$, Assumption 4.10 and the condition $\rho(f(\mathcal{A}_{\text{inc}})) = \rho(f(\mathcal{S}))$ imply that $|\mathcal{A}_{\text{inc}} \cap \mathcal{I}|_\varphi = |\mathcal{S} \cap \mathcal{I}|_\varphi$. Note that $\mathcal{S} \subseteq \mathcal{A}_{\text{inc}}$ implies $|\mathcal{S} \cap \mathcal{I}|_\varphi \leq |\mathcal{A}_{\text{inc}} \cap \mathcal{I}|_\varphi$, with the equality being achieved when $\mathcal{S} \cap \mathcal{I} = \mathcal{A}_{\text{inc}} \cap \mathcal{I}$. Therefore, $\mathcal{S} \cap \mathcal{I} = \mathcal{A}_{\text{inc}} \cap \mathcal{I}$. Then, the minimization problem $\min_{\mathcal{S} \in \mathcal{S}_v(\mathcal{A}_{\text{inc}})} |\mathcal{S}|_\eta =$

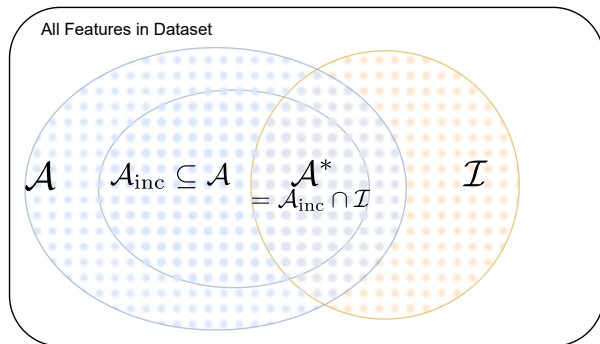

Figure 5: **Soundness evaluation.** Computing the soundness of $\mathcal{A}$ in a single step is unfeasible. Instead, we incrementally include a subset $\mathcal{A}_{\text{inc}}$ in input and compute its soundness. This process involves identifying the optimal set $\mathcal{A}^*$ and calculating $\frac{|\mathcal{A}^*|_\eta}{|\mathcal{A}_{\text{inc}}|_\eta}$. A particular $\mathcal{A}^*$ is associated with a specific predictive level (i.e., model performance). When comparing two attribution methods, we can standardize the predictive level, allowing us to evaluate the soundness at this fixed level.

$\min_{\mathcal{S} \in \mathcal{S}_v(\mathcal{A}_{\text{inc}})}(|\mathcal{S} \cap \mathcal{I}|_\eta + |\mathcal{S} \cap (\mathcal{F} \setminus \mathcal{I})|_\eta)$ boils down to $\min_{\mathcal{S} \in \mathcal{S}_v(\mathcal{A}_{\text{inc}})} |\mathcal{S} \cap (\mathcal{F} \setminus \mathcal{I})|_\eta$, because $\mathcal{S} \cap \mathcal{I} = \mathcal{A}_{\text{inc}} \cap \mathcal{I}$ implies $|\mathcal{S} \cap \mathcal{I}|_\eta = |\mathcal{A}_{\text{inc}} \cap \mathcal{I}|_\eta$, which is a constant with fixed $\mathcal{A}_{\text{inc}}$. Thus, according to the definition of $\mathcal{A}^*$ and $\mathcal{A}_{\text{inc}} \cap \mathcal{I} \neq \emptyset$, we have $|\mathcal{A}^* \cap (\mathcal{F} \setminus \mathcal{I})|_\eta = 0$ and $|\mathcal{A}^*|_\eta = |\mathcal{A}_{\text{inc}} \cap \mathcal{I}|_\eta + |\mathcal{A}^* \cap (\mathcal{F} \setminus \mathcal{I})|_\eta = |\mathcal{A}_{\text{inc}} \cap \mathcal{I}|_\eta$. Therefore, the soundness of $\mathcal{A}_{\text{inc}}$ is $\frac{|\mathcal{A}_{\text{inc}} \cap \mathcal{I}|_\eta}{|\mathcal{A}_{\text{inc}}|_\eta} = \frac{|\mathcal{A}^*|_\eta}{|\mathcal{A}_{\text{inc}}|_\eta}$. We can also prove that our minimizer $\mathcal{A}^*$ is the set that contains all predictive information of $\mathcal{A}_{\text{inc}}$. $|\mathcal{A}^* \cap (\mathcal{F} \setminus \mathcal{I})|_\eta = 0$ indicates $\mathcal{A}^* \subseteq \mathcal{I}$ since $\eta(F) > 0$ holds for all $F \in \mathcal{A}^* \subseteq \mathcal{A}$. Next, $\mathcal{A}^* \subseteq \mathcal{I}$ and $\mathcal{A}^* \subseteq \mathcal{A}_{\text{inc}}$ yields $\mathcal{A}^* \subseteq (\mathcal{A}_{\text{inc}} \cap \mathcal{I})$. Combining with $|\mathcal{A}^*|_\eta = |\mathcal{A}_{\text{inc}} \cap \mathcal{I}|_\eta$, we have $\mathcal{A}^* = \mathcal{A}_{\text{inc}} \cap \mathcal{I}$. □

Theorem 4.11 shows that the soundness of $\mathcal{A}_{\text{inc}}$ can be computed by finding an element from $\mathcal{S}_v(\mathcal{A}_{\text{inc}})$ that has the minimum attribution. Figure 5 depicts the relationship between $\mathcal{A}^*$, $\mathcal{A}_{\text{inc}}$ and $\mathcal{I}$. Additionally, it is crucial to recognize that our definition of $\mathcal{A}_{\text{inc}}$ satisfies the conditions in Theorem 4.11, specifically $\mathcal{A}_{\text{inc}} \subseteq \mathcal{A}$ and $\mathcal{A}_{\text{inc}} \cap \mathcal{I} \neq \emptyset$. This inequality holds because $\rho(f(\mathcal{A}_{\text{inc}})) > 0$, and Assumption 4.10 implies $|\mathcal{A}_{\text{inc}} \cap \mathcal{I}|_\varphi > |\emptyset|_\varphi$.

To facilitate comparison between different attribution methods, we can compute and compare their soundness at a fixed predictive level $v$. Algorithm 1 shows how to compute the soundness at predictive level $v$. We gradually include features with the highest attribution values in the input. During the set expansion of $\mathcal{A}_{\text{inc}}$, we simultaneously perform minimization as shown in Theorem 4.11 by examining and excluding non-predictive features based on the change in the model's performance. After reaching predictive level $v$, we can directly calculate soundness based on the set $\mathcal{A}_{\text{inc}}$ and the optimized set $\mathcal{A}^*$. Nonetheless, Algorithm 1 only approximates the actual soundness. The iterative algorithm uses accuracy to identify informative features, which may introduce bias, as certain features could be more contributive in conjunction with different sets of features. To minimize this bias, our approach incorporates a batch of features rather than a single feature at each expansion step. Moreover, we sequentially include features from highest to lowest attribution, halting at a specific accuracy threshold before saturation to capture potentially important features. A comprehensive description of the soundness evaluation algorithm is provided in Appendix E. Notably, linear imputation (Rong et al., 2022) is used in soundness (and completeness) evaluation procedures to mitigate OOD effects caused by feature removal (as we progressively include a portion of features). It is crucial to recognize that the soundness evaluation differs fundamentally from classical Insertion/Deletion metrics. While soundness evaluates the ratio of attribution values associated with two feature sets, Insertion/Deletion assesses accuracy following feature removal and employs attribution values solely for feature sorting.

## 4.3 Completeness evaluation

Completeness, as previously discussed, assesses the extent to which all predictive features are detected within attribution maps. Per Definition 4.9, removing attributed features from a complete attribution map might

---

**Algorithm 2** Completeness evaluation at attribution threshold $t$

---

1: **Input:** dataset $\mathbb{D} = \{x^{(i)}, y^{(i)}\}_{i=1}^{N}$ with attribution maps $\{\mathcal{A}^{(i)}\}_{i=1}^{N}$, model $f$, attribution threshold $t$.
2: **Initialize** $s_0 = \text{Accuracy}(f, \mathbb{D})$;
3: $\{\mathcal{A}_{>t}^{(i)}\}_{i=1}^{N} = \text{Threshold}(\{\mathcal{A}^{(i)}\}_{i=1}^{N}, t)$;          // Include features w/ attr. val. > t.
4: $\{\mathcal{A}_{\leq t}^{(i)}\}_{i=1}^{N} = \{\mathcal{F} \setminus \mathcal{A}_{>t}^{(i)}\}_{i=1}^{N}$;
5: $\tilde{\mathbb{D}} = \{\mathcal{A}_{\leq t}^{(i)}, y^{(i)}\}_{i=1}^{N}$;
6: **Return** $\Delta s_t = s_0 - \text{Accuracy}(f, \tilde{\mathbb{D}})$          // compare $\Delta s_t$ for completeness comparison.

---

also remove predictive features. However, if the attribution method has low completeness, removing the attributed features would not eliminate all predictive features in the input. Theorem 4.12 tells us how to compare the completeness using the remaining features after the removal of attributed features.

**Theorem 4.12.** *Let $\mathcal{A}_1$ and $\mathcal{A}_2$ be the attributed features given by two attribution methods, respectively. If $\rho(f(\mathcal{F} \setminus \mathcal{A}_1)) < \rho(f(\mathcal{F} \setminus \mathcal{A}_2))$, then the attribution method associated with $\mathcal{A}_1$ is more complete than the one associated with $\mathcal{A}_2$.*

**Proof**: The condition of $\rho(f(\mathcal{F} \setminus \mathcal{A}_1)) < \rho(f(\mathcal{F} \setminus \mathcal{A}_2))$ together with Assumption 4.10 implies that $|\mathcal{I} \cap (\mathcal{F} \setminus \mathcal{A}_1)|_\varphi < |\mathcal{I} \cap (\mathcal{F} \setminus \mathcal{A}_2)|_\varphi$. Here, for any set $\mathcal{S}$, $|\mathcal{I} \cap (\mathcal{F} \setminus \mathcal{S})|_\varphi = \sum_{F \in (\mathcal{I} \cap (\mathcal{F} \setminus \mathcal{S}))} \varphi(F) = \sum_{F \in \mathcal{I}} \varphi(F) - \sum_{F \in (\mathcal{I} \cap \mathcal{S})} \varphi(F) = |\mathcal{I}|_\varphi - |\mathcal{I} \cap \mathcal{S}|_\varphi$. Using this for $\mathcal{S} = \mathcal{A}_1$ and $\mathcal{A}_2$, we have that $|\mathcal{I} \cap (\mathcal{F} \setminus \mathcal{A}_1)|_\varphi < |\mathcal{I} \cap (\mathcal{F} \setminus \mathcal{A}_2)|_\varphi \Leftrightarrow |\mathcal{I}|_\varphi - |\mathcal{I} \cap \mathcal{A}_1|_\varphi < |\mathcal{I}|_\varphi - |\mathcal{I} \cap \mathcal{A}_2|_\varphi \Leftrightarrow \frac{|\mathcal{I} \cap \mathcal{A}_2|_\varphi}{|\mathcal{I}|_\varphi} < \frac{|\mathcal{I} \cap \mathcal{A}_1|_\varphi}{|\mathcal{I}|_\varphi}$. $\square$

Based on the above analysis, we present Algorithm 2 for evaluating completeness at an attribution threshold $t$. We start by removing input features with attribution values above $t$. Then we pass the remaining features along with imputed features to the model and report the difference in the model performance between the original and the remaining features. A higher score difference means higher completeness. The detailed procedure of completeness evaluation is shown in Appendix F. The completeness evaluation diverges from the Deletion/Insertion approach in its method of feature removal: it removes features with attribution exceeding a specific value, whereas Insertion/Deletion removes features whose ranking is better than a certain threshold (e.g. top 20%). Despite the subtlety of this distinction, as discussed in Section 5.2, the completeness evaluation is capable of discerning differences in attribution values, a nuance that Insertion/Deletion may fail to capture.

# 5 Experiments

In this section, we begin by validating our proposed metrics in a controlled setting, serving as a sanity check. We subsequently underscore the importance of considering attribution values during evaluation, rather than just focusing on feature ranking order. This allows for differentiation between methods that rank features identically but assign differing attribution values. Lastly, we further demonstrate that using our two metrics together provides a more fine-grained evaluation, enabling us to gain a deeper understanding of how an attribution method can be improved.

## 5.1 Validation of the proposed metrics

In this section, we first validate whether the metrics work as expected and reflect the soundness and completeness properties. In other words we evaluate whether the proposed algorithms follow the predictions of our theories. We empirically validate the soundness and completeness metrics using a synthetic setting. Through a designed synthetic dataset and a transparent linear model, we obtain ground truth attribution maps that are inherently sound and complete. These inherently sound and complete attribution maps are then modified to probe expected effects in completeness or soundness, allowing us to test how our proposed metrics behave in different situations. By increasing the attribution values of non-predictive features, we introduce extra attribution (termed as *Introduce*) which hurts soundness but improves completeness. Conversely, removing attribution (denoted as *Remove*) lowers completeness without influencing soundness. Our

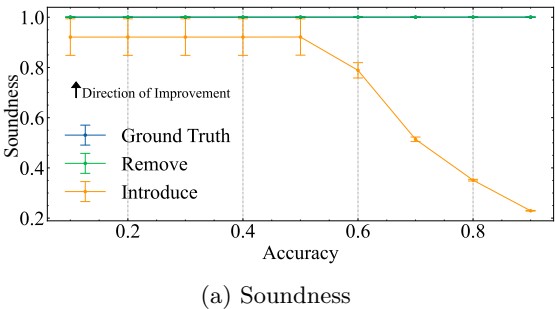 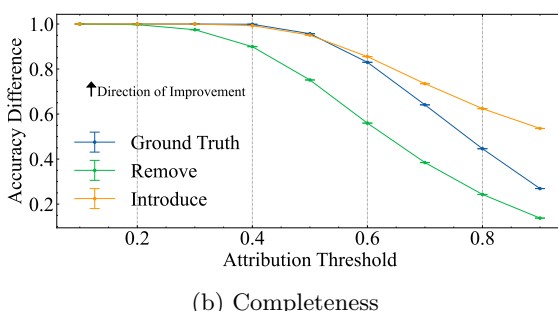

(a) Soundness  (b) Completeness

Figure 6: **Completeness and soundness evaluations on the synthetic dataset.** The curve and error bar respectively represent the mean and variance across 1000 trials. The curves of Remove and Ground Truth are overlapped in (a) as they both saturate at 1. Both soundness and completeness evaluations can reliably reflect the modifications in the attribution maps.

objective is to evaluate these modified attribution maps to ensure our proposed metrics accurately capture changes in both soundness and completeness.

The synthetic two-class dataset consists of data points sampled from a 200-dimensional Gaussian $\mathcal{N}(\mathbf{0}, \mathbf{I})$. Data points are labeled based on the sign of the sum of their features. Let $x_i$ be the $i$-th input feature and $\sigma(\cdot)$ is a step function that rises at 0. The linear model, formulated as $y = \sigma(\sum_i x_i)$, is designed to replicate the data generation process and is transparent, allowing us to obtain sound and complete ground truth attribution maps. Appendix G.1 provides further details. We randomly add and remove attribution from ground truth feature maps 1000 times each. Then, we compare the soundness and completeness between modified and original attribution maps.

Statistical results in Figure 6 show that *Remove* consistently outperforms ground truth attribution in Completeness, whereas *Introduce* underperforms ground truth attribution in Completeness. In Soundness evaluation, the ranking of the three methods inverses. Note that the optimum soundness of ground truth attribution is 1, which can be also reached by *Remove*. In conclusion, the evaluations behave as expected, validating our proposed metrics in this case.

## 5.2 Comparison with order-based metrics

Attribution methods aim to determine the contribution values of features beyond merely ranking them by importance. Consequently, evaluating these methods necessitates consideration of the actual attribution values. Both Completeness and Soundness metrics incorporate attribution values: the former uses value-based thresholds for feature removal, while the latter, denoted as $\frac{|\mathcal{A}^*|_\eta}{|\mathcal{A}_{\text{inc}}|_\eta}$, inherently captures variations in attribution values. Next, we show that this consideration of attribution values results in a more refined evaluation.

For the following experiments, we employ a VGG16 (Simonyan & Zisserman, 2015) pre-trained on ImageNet (Deng et al., 2009) and conduct feature attribution on the ImageNet validation set. We apply the *Remove* and *Introduce* modifications to the original attribution maps produced by a given attribution method, such as GradCAM, as visualized in Figure 7a. These modifications are intentionally designed to slightly adjust the ordering of attributions, yet they significantly alter the attribution values. Consequently, the original attribution maps and those modified by *Remove* and *Introduce* are differentiated not just in terms of attribution values but also in their visual presentation, as illustrated in Figure 7a. A well-designed evaluation metric must be capable of capturing these differences clearly. Therefore, for a metric to be considered effective, the curves representing the evaluation results for the original, *Remove*-, and *Introduce*-modified attribution maps should be distinct and *non-overlapping*.

As illustrated in Figure 7, the curves representing *Remove* and *Introduce* overlap in ROAD and Deletion. This is attributed to the fact that these metrics are based solely on the order of attribution, which remains nearly unchanged between *Remove* and *Introduce*-modified attribution maps. In contrast, the curves for the

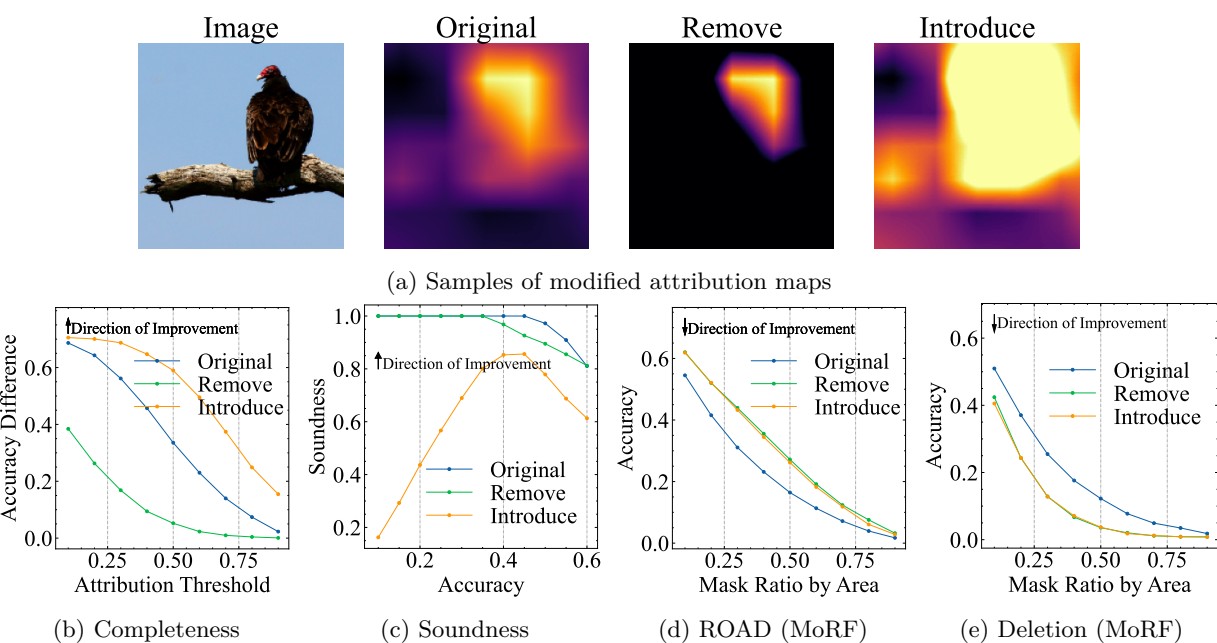

(a) Samples of modified attribution maps

(b) Completeness  (c) Soundness  (d) ROAD (MoRF)  (e) Deletion (MoRF)

Figure 7: **Analysis of our metrics and order-based metrics.** (a) Modified attribution maps. The modifications result in only minimal changes to the feature order. These modified maps are noticeably different from the original. An evaluation metric should capture this distinctiveness. By taking attribution values into account, Completeness (b) and Soundness (c) aptly distinguish the modifications in the attribution maps. Conversely, the differences between curves are less obvious in ROAD (d) and Deletion (e). A side note on (c) is that *Remove* might not always preserve soundness. This is because original attribution maps are not always the same as ground truth maps (inaccessible in the real world), and *Remove* can eliminate both predictive and non-predictive features.

value-sensitive metrics Completeness and Soundness are noticeably distinct, highlighting their sensitivity to changes in attribution values. To quantitatively assess the differences between evaluation curves, we calculate the *minimal* Hausdorff distance between pairs of curves, denoted as $\min_{p,q}$ Hausdorff$(p,q)$, where $p,q \in \{$Original, *Introduce*, *Remove*$\}$. A minimal Hausdorff distance approaching zero signifies an overlap in the evaluation results, indicating that the metric fails to distinguish between the modified and original attribution maps.

We implement three pairs of different modification schemes for *Introduce* and *Remove*, which are elaborated in Appendix G. These modification schemes were applied to attribution maps generated by GradCAM, IG, and ExPerturb, and the process of modification and evaluation was iterated for each scheme. The resulting minimal Hausdorff distances were then averaged. As indicated in Table 1, the Hausdorff distances for Completeness and Soundness metrics are significantly greater than zero. This demonstrates that these metrics can effectively differentiate between the modified and original attribution maps, even when the changes in attribution order are minimal. Conversely, the order-based metrics, ROAD and Deletion, demonstrate overlaps in their curves, indicating their inadequacy in discerning subtle distinctions. This contrast highlights the superior sensitivity of Completeness and Soundness metrics in evaluating the nuances of attribution maps.

## 5.3  Benchmark experiments

As previously discussed, the misalignment between attributed features and predictive features arises from two types of attribution errors. By employing both the Completeness and Soundness metrics, we can identify which type of error reduction contributes to the superior performance of one method.

Table 1: Minimal Hausdorff distances across evaluation curves for Original, *Remove-*, and *Introduce*-modified attribution maps. Near-zero distances in ROAD and Deletion imply curve overlap, indicating limited differentiation between the attribution maps. Conversely, significant distances in Completeness and Soundness highlight their ability to distinctively evaluate and differentiate attribution maps, showcasing their finer granularity.

| Metric | Completeness | Soundness | Deletion | ROAD |
|---|---|---|---|---|
| Hausdorff Distance | $0.503 \pm 0.112$ | $0.183 \pm 0.046$ | $0.019 \pm 0.011$ | $0.014 \pm 0.009$ |

(a) Soundness     (b) Completeness     (c) ROAD     (d) Deletion

Figure 8: **Benchmark of IG ensembles.** By employing both the Completeness and Soundness metrics, we can see that the superiority of ensemble methods over IG is predominantly in their soundness.

**Benchmark of ensemble methods** Several ensemble methods have been proposed as a means to improve attribution methods. In this study, we focus specifically on three ensembles of IG: SmoothGrad (IG-SG) (Smilkov et al., 2017), SmoothGrad$^2$ (IG-SQ) (Hooker et al., 2019), and VarGrad (IG-Var) (Adebayo et al., 2018). Intriguingly, only IG-SG displays an enhancement in completeness (Figure 8b), consistent with visual results from earlier studies. We present supplementary visual results in Appendix G.4 for further scrutiny. Previous research (Smilkov et al., 2017) has observed that gradients can fluctuate significantly in neighboring samples. Consequently, the aggregation of attribution from neighboring samples can mitigate false attribution—specifically those arising from non-predictive features receiving attribution—and notably enhance soundness, as depicted in Figure 8a. However, the benefits of ensemble methods are not so clear in ROAD (Figure 8c) or Deletion (Figure 8d).

**Benchmark of various attribution methods** We conduct a comparative analysis of multiple attribution methods using our metrics with the goal of guiding the selection of suitable methods for diverse applications. As illustrated in Figure 9, most of the evaluated methods excel in one metric over the other, suggesting their suitability varies based on specific scenarios. For applications like clinical medicine, where capturing all relevant features is essential, methods with higher completeness, like ExPerturb, stand out. On the other hand, in situations where falsely identifying non-predictive features as significant could be detrimental, methods showcasing superior soundness, such as IBA or GradCAM, are preferable.

## 6 Conclusion and limitations

In this paper, we first revealed the potential pitfalls in existing faithfulness evaluation of attribution methods. Subsequently, we defined two important properties of attribution: soundness and completeness. We also proposed methodologies for measuring and comparing them. The two metrics work in conjunction and offer a higher level of differentiation granularity. Empirical validation convincingly demonstrated the effectiveness of our proposed metrics. Furthermore, we undertook a benchmark of ensemble methods, revealing that these methods can considerably improve the soundness of the baseline. Lastly, we extended the comparative analysis to a broader range of attribution methods to provide guidance for selecting methods for different

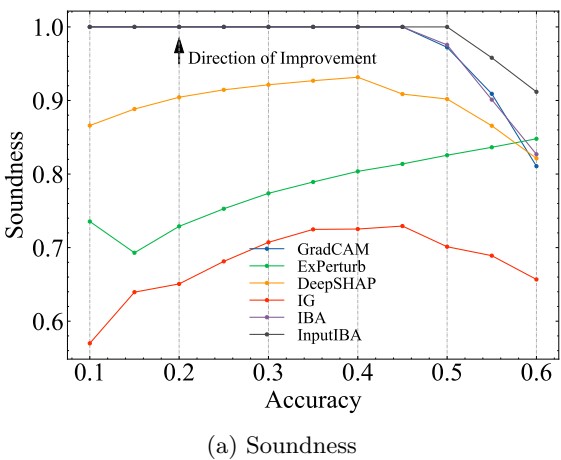
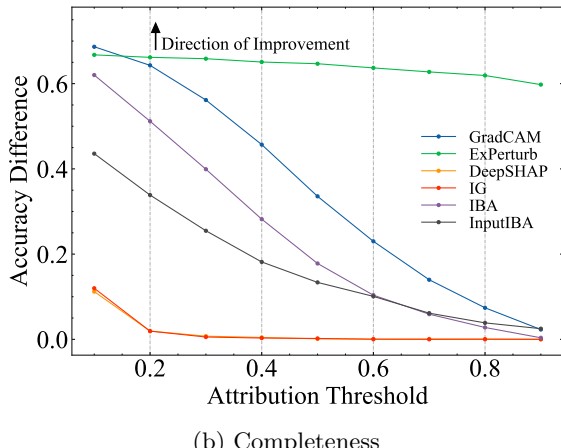

(a) Soundness

(b) Completeness

Figure 9: **Benchmark of different methods.** Although no method exhibits superior performance in both Completeness and Soundness, some of them perform well in one of these metrics, implying their suitability for applications which have high demand for the corresponding property.

practical applications. One limitation of our evaluations is that their efficacy hinges on the assessment of model performance. Accuracy may not be the appropriate performance metric in some cases. Therefore, additional research in the future is needed to find better performance indicators for different tasks. In addition, Theorem 4.11 does not limit the selection of $\mathcal{A}_{\text{inc}}$, and better set expansion strategies for $\mathcal{A}_{\text{inc}}$ could yield more precise evaluation outcomes.

## 7 Acknowledgment

This research is partially supported by the National Research Foundation Singapore under the AI Singapore Programme (AISG Award No: AISG2-TC-2023-010-SGIL) and the Singapore Ministry of Education Academic Research Fund Tier 1 (Award No: T1 251RES2207), and UKRI grant: Turing AI Fellowship EP/W002981/1.

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

## A  Notations

Table 2 summarizes the notations used in this paper.

Table 2: Table of notations.

| | |
|---|---|
| $v$ | a predictive level (i.e. a specific level of model performance) |
| $f$ | a model to be explained |
| $\rho$ | a performance metric to assess the performance of model $f$ |
| $\mathbb{D}$ | a (labeled) dataset |
| $\mathbb{D}^{(i)}_{\mathrm{P,Train}}$ | a perturbed training set using perturbation strategy $i$ |
| $\mathbb{D}^{(i)}_{\mathrm{P,Test}}$ | a perturbed test set using perturbation strategy $i$ |
| $\mathbb{D}^{(i)}_{\mathrm{S}}$ | a semi-natural dataset constructed by modifying the original dataset using modification method $i$ |
| $\mathcal{F}$ | a set that contains all features in the dataset |
| $\mathcal{A}$ | a set of attributed features |
| $\mathcal{I}$ | a set of predictive features for the model |
| $\lvert \cdot \rvert_\eta$ | the operator to calculate the sum of attribution |
| $\lvert \cdot \rvert_\varphi$ | the operator to calculate the sum of class-related information |
| $F$ | a single feature in $\mathcal{F}$ |
| $\varphi(F)$ | a function that returns the information value of a feature $F$ |
| $\eta(F)$ | a function that returns the attribution value of a feature $F$ |
| $\mathcal{A}_{\mathrm{inc}}$ | a subset of the most salient features that reach $\rho(f(\mathcal{A}_{\mathrm{inc}})) = v > 0$ |
| $\mathcal{S}_v(\mathcal{A}_{\mathrm{inc}})$ | for a given set $\mathcal{A}_{\mathrm{inc}}$, we define $\mathcal{S}_v(\mathcal{A}_{\mathrm{inc}}) = \{\mathcal{S} \subseteq \mathcal{A}_{\mathrm{inc}} : \rho(f(\mathcal{A}_{\mathrm{inc}})) = \rho(f(\mathcal{S}))\}$ |
| $\mathcal{A}^*$ | minimizer of $\min_{\mathcal{S} \in \mathcal{S}_v(\mathcal{A}_{\mathrm{inc}})} \lvert \mathcal{S} \rvert_\eta$ |

## B  Broader Impacts

We believe that our proposed completeness and soundness evaluations open up many innovative directions. For instance, we have shown that the ensemble methods can greatly enhance the soundness of baselines such as IG and DeepSHAP. However, the gain in completeness is very marginal. It would be interesting to investigate how to also improve the completeness of IG, DeepSHAP, or their ensembles. In addition, Extremal Perturbations demonstrate lower soundness than IBA and GradCAM that perform attribution on the hidden neurons. This might suggest that the semantic information in hidden layers can be utilized in the optimization process of the Extremal Perturbations to reduce false attribution.

## C  Additional experiments for revealing the issues with retraining-based metrics

In this section, we report an additional experiment to further illustrate the issue with retraining-based evaluation. For this additional experiment, we use the CIFAR-10 (Krizhevsky et al., 2009a) dataset. The model is a tiny ResNet (He et al., 2016) with only 8 residual blocks. Training is conducted using Adam (Kingma & Ba, 2015) optimizer with a learning rate of 0.001 and weight decay of 0.0001. The batch size used for the training is 256, and we train a model in 35 epochs. Next, we describe how to construct the maliciously modified dataset for retraining.

In the retraining experiment shown in Figure 10, we generate a modified dataset from the original CIFAR-10 dataset. In this additional experiment, we only perturb 5% of each training image and replace the perturbed pixels with black pixels. The perturbation is correlated with class labels. For different classes, we select different positions close to the edge of the image so the object (usually at the center of the image) is barely removed.

The result is shown in Figure 10. We summarize our findings in the caption of Figure 10.

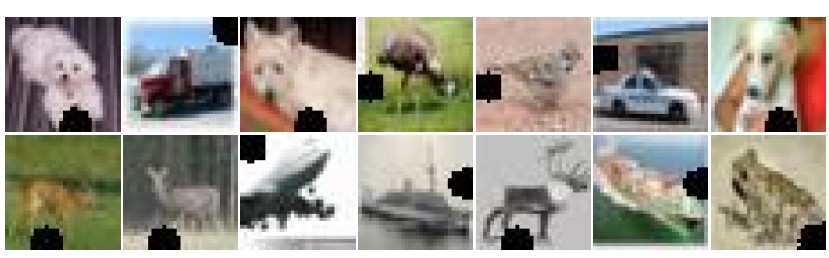

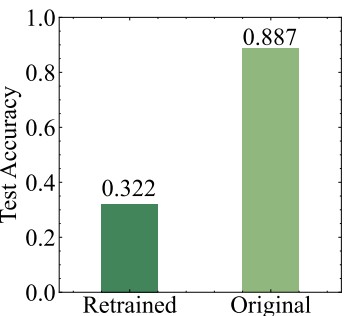

(a) Sample training images

(b) Evaluation on unperturbed test set

Figure 10: Retrain on the perturbed dataset with spurious correlation. (a) illustrates sample images from the perturbed training set. Only a small portion of pixels on the edge is removed. Hence, the class object is usually intact after perturbation. However, the position of the removed region depends on the label of the image. (b) Test accuracy on the unperturbed test set. Although objects are not removed in the perturbed training set, the retrained model achieves much lower test accuracy on the original test set than the model trained on the original dataset. This means that the retrained model ignores the object but learns to perform classification based on the spurious correlation introduced by perturbation. We would like to further demonstrate the issue of retraining, that the retrained model fails to learn exclusively from remaining features in the perturbed dataset. Hence, we cannot use the model performance to measure the information loss caused by perturbation.

# D   Additional Experiments and Experiment Configurations on Semi-natural Datasets

## D.1   Evaluation with Models Retrained on CIFAR-100, Semi-natural, and Pure Synthetic Datasets

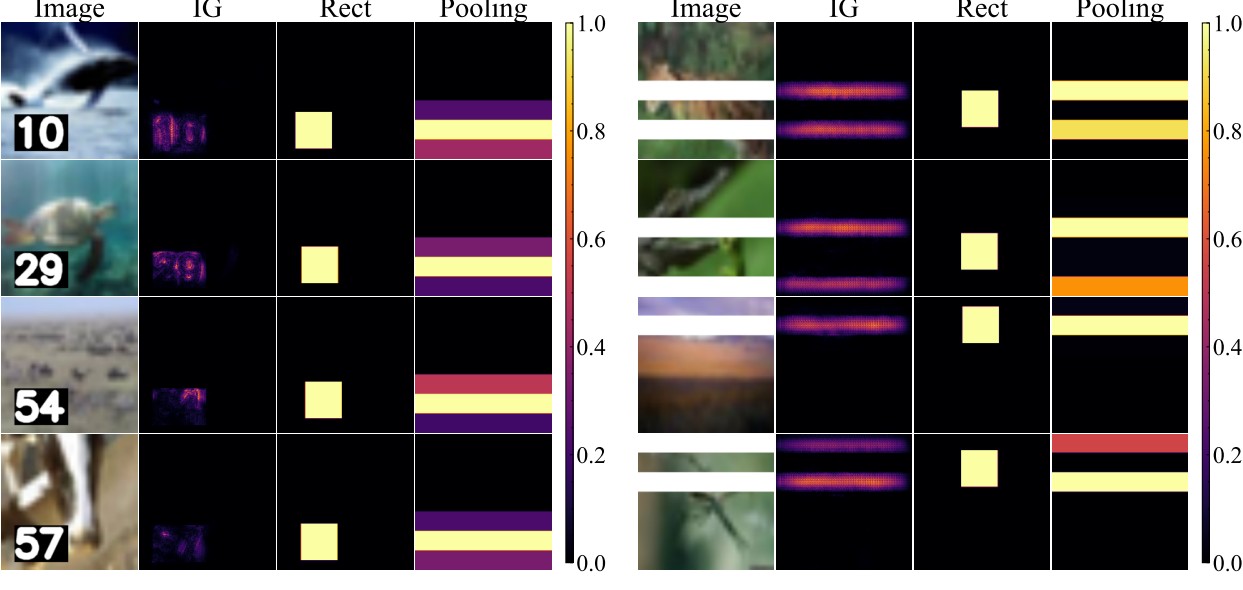

(a) Semi-natural dataset with number watermarks (i.e. $\mathbb{D}_S^{(1)}$)

(b) Semi-natural dataset with stripe watermarks (i.e. $\mathbb{D}_S^{(2)}$)

Figure 11: Semi-natural datasets with different watermarks. Two types of attribution maps are crafted based on IG attribution maps by utilizing prior knowledge about the watermarks. *Rect* is designed to fit $\mathbb{D}_S^{(1)}$, while *Pooling* is designed to fit $\mathbb{D}_S^{(2)}$.

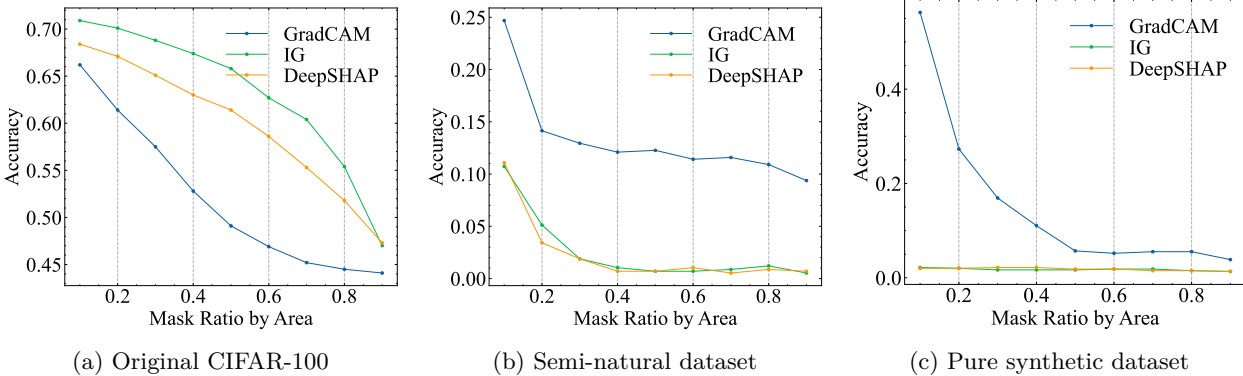

(a) Original CIFAR-100        (b) Semi-natural dataset        (c) Pure synthetic dataset

Figure 12: ROAD evaluation on different datasets. The performance of attribution methods is very distinct across different datasets. We observe that the ROAD result on the original dataset is different from semi-natural dataset and pure synthetic dataset. However, the ROAD result on the semi-natural dataset is very similar to the result on the pure synthetic dataset. This implies that the semi-natural dataset changes the implicit learning task from learning representations of the original images to learning representations of synthetic symbols introduced during dataset construction. Subsequently, the task is greatly simplified and substantially divergent from the real dataset.

In Section 3.2, we argue that the attribution methods can behave much differently when explaining the model retrained on a semi-natural dataset. As a result, it is not faithful to use the evaluation result on the semi-natural dataset as an assessment for the feature attribution methods. In this section, we demonstrate this issue with an experiment.

We first show the datasets used in the experiment. we re-assign the labels for CIFAR-100 (Krizhevsky et al., 2009b) images as suggested in (Zhou et al., 2022). Next, we inject two types of watermarks into the images and a blank canvas, obtaining two pairs of semi-natural and pure synthetic datasets, respectively. **Note that the semi-natural datasets are also used in Section 3.2**. The watermarks are designed as follows:

- **Number watermark:** as depicted in Figure 11a left, we first insert a black rectangular region in the image and then put the white number sign within the black region.

- **Stripe watermark:** as depicted in Figure 11b left, we first encode the label into a 7-digit binary number and divide the image into 7 equal-height regions. Next, we set the pixels in each region to 255 if the corresponding digit is 1; otherwise, we leave the pixels unchanged.

The following experiment is conducted on the semi-natural and pure synthetic dataset with *number watermarks*. We first train a VGG-16 on the CIFAR-100, semi-natural, and pure synthetic datasets, respectively. After obtaining the classifiers, we apply GradCAM, IG, and DeepSHAP to them to get the attribution maps. In the end, we benchmark the three attribution methods on each dataset using ROAD (Rong et al., 2022).

The models achieve 70.4% on CIFAR-100, 99.4% on the semi-natural dataset, and 99.9% on the pure synthetic dataset, respectively. The difference in accuracy shows that the learning tasks are differently complex across three datasets. Furthermore, as depicted in Figure 12, GradCAM outperforms IG and DeepSHAP on CIFAR-100, while IG and DeepSHAP are much better than GradCAM on the semi-natural and pure synthetic datasets. *The evaluation results on the semi-natural dataset cannot correctly reflect attribution methods' performance on the real-world dataset.*

### D.2 Details on Designing *Rect* and *Pooling* Attribution Maps

In this section, we show how we design attribution maps by leveraging prior knowledge about semi-natural datasets. The target is to craft two types of attribution maps, with one performing well on the semi-natural dataset with number watermarks (i.e., $\mathbb{D}_S^{(1)}$ in Section 3.2) and another performing well on the semi-natural

dataset with stripe watermarks (i.e., $\mathbb{D}_S^{(2)}$ in Section 3.2). Both attribution maps are generated based on IG attribution maps. The following are the design details:

- *Rect* is designed to fit $\mathbb{D}_S^{(1)}$, where the modified pixels (i.e., *Effective Region* in (Zhou et al., 2022)) are the black rectangular region (where the numbers are located) that we injected into an image. To craft attribution maps, we can also put a rectangular region full of value 1.0 on a background of value 0.0. The question is where to put such a rectangular region. For each IG attribution map, we average across the pixels' spatial locations using the attribution values as weights, obtaining a "weighted center" of the attribution map. Then, we put the rectangular region of spatial size $60 \times 60$ at the weighted center. More samples are shown in Figure 11a.

- *Pooling* is designed to fit $\mathbb{D}_S^{(2)}$, where the modified pixels are the equal-height regions associated with digit 1. After knowing the shape of watermarks, we can design attribution maps composed of 7 equal-height regions. To do so, we apply average pooling to each IG attribution map, obtaining a 7-element attribution vector. Next, we fill each region in the crafted attribution map with the corresponding value in the attribution vector. More samples are shown in Figure 11b.

## E    Implementation Details of Soundness Metric

Values in attribution maps are usually continuous. Hence, it is possible that an attribution method only has satisfactory performance only in a certain attribution value interval. To evaluate the overall performance of an attribution method, we use Algorithm 1 as a basic building block to establishing the progressive evaluation procedure. Specifically, we evaluate soundness at different predictive levels indicated by the model performance. How soundness is calculated at specific predictive level has been explained in Section 4.2.

Since the attribution method considers features with higher attribution values to be more influential for the model decision-making, we expand our evaluation set by gradually including the most salient features that are not yet in the evaluation set. As shown in Algorithm 3, the expansion of the evaluation set happens by decreasing the mask ratio. For the soundness metric, the mask ratio $v$ means that the top $v$ pixels in an attribution map sorted in ascending order are masked (i.e., area-based LeRF). We start from $v = 0.98$ and decrease $v$ by the step size of 0.01. This is equivalent to first inserting 2% of the most important pixels in a blank canvas and inserting 1% more pixels at each step. If the accuracy difference between the current step and the previous step is smaller than the threshold 0.01, then the attribution of newly added pixels is deemed to be false attribution and will be discarded. Algorithm 3 demonstrates a more detailed procedure compared to Algorithm 1 in Section 4.2. Note that some notations are overloaded.

## F    Implementation Details of Completeness Metric

To obtain the overall completeness performance of an attribution method, we again select subsets of an attribution set and evaluate the completeness of these subsets.

Algorithm 4 demonstrates the full computation process. For the completeness metric, the attribution threshold $t$ means that the pixels with attribution between $[t, 1]$ will be masked (i.e., value-based MoRF). We start from $v = 0.9$ and decrease $t$ by the step size of 0.1. Compared to Algorithm 2 in Section 4.3, the pseudo-code in Algorithm 4 is more detailed and closer to the actual implementation. Note that some annotations are overloaded.

## G    Validation and Benchmark Experiments

### G.1    Validation Tests

We create a two-class dataset of 1000 sample data points, and each data point has 200 features. The data point is sampled from a 200-dimensional $\mathcal{N}(\mathbf{0}, \mathbf{I})$ Gaussian distribution. If all features for a sample point sum up to be greater than zero, we assign a positive class label to this sample. Otherwise, a negative class label is

---

**Algorithm 3** Soundness evaluation with accuracy ($s_m$) as performance indicator

---

1: **Input**: $f$: model; $\mathbb{D} = \{x^{(i)}, y^{(i)}\}_{i=1}^{N}$: labeled dataset with attribution maps $\{\mathcal{A}^{(i)}\}_{i=1}^{N}$; $\phi$: perturbation function; $\psi$: noisy linear imputation function; $\mathbb{M} = \{0.99, 0.98, 0.97, \ldots 0.01\}$: mask ratios (by area); Accuracy: accuracy evaluation function. $\epsilon$: accuracy threshold; NewAdded: function that identifies the included features at the current step and newly added features compared to the last step.

2: **Output**: $\mathbb{P}$: A list of tuples with each tuple being the accuracy and the soundness value.

3: Initialize $\mathbb{P} \leftarrow [\,]$;

4: $\{\hat{\mathcal{A}}^{(i)}\}_{i=1}^{N} \leftarrow \{\emptyset\}_{1}^{N}$; // Initialize the set of features with false attribution.

5: $m_0 \leftarrow 1$; // Initialize the mask ratio at the previous step.

6: $s_0 \leftarrow 0$; // Initialize the accuracy at the previous step.

7: **for** $m$ in $\mathbb{M}$ **do**

8:    $\tilde{\mathbb{D}}_m \leftarrow \emptyset$   // Initialize imputed dataset

9:    **for** $(x^{(i)}, y^{(i)}), \mathcal{A}^{(i)}$ in $(\mathbb{D}, \{\mathcal{A}^{(i)}\}_{i=1}^{N})$ **do**

10:       $\hat{x}^{(i)} \leftarrow \phi(x^{(i)}, \mathcal{A}^{(i)}, m)$;   // Perturb image in LeRF order.

11:       $\tilde{x}^{(i)} \leftarrow \psi(\hat{x}^{(i)})$;   // Impute image.

12:       $\mathcal{A}_{\text{inc}}^{(i)}, \Delta\mathcal{A}^{(i)} \leftarrow \text{NewAdded}(\mathcal{A}^{(i)}, m, m_0)$; // Identify included features at the current step and newly added feature compared to the last step.

13:       $\text{Append}(\tilde{\mathbb{D}}_m, (\tilde{x}^{(i)}, y^{(i)}, \mathcal{A}_{\text{inc}}^{(i)}, \Delta\mathcal{A}^{(i)}))$;

14:    **end for**

15:    $s_m \leftarrow \text{Accuracy}(f, \tilde{\mathbb{D}}_m)$;   Accuracy on the imputed dataset.

16:    **if** $s_m - s_0 < \epsilon$ **then**

17:       **for** $((\tilde{x}^{(i)}, y^{(i)}, \mathcal{A}_{\text{inc}}^{(i)}, \Delta\mathcal{A}^{(i)}), \hat{\mathcal{A}}^{(i)})$ in $(\tilde{\mathbb{D}}_m, \{\hat{\mathcal{A}}^{(i)}\}_{i=1}^{N})$ **do**

18:          $\hat{\mathcal{A}}^{(i)} \leftarrow \hat{\mathcal{A}}^{(i)} \cup \Delta\mathcal{A}^{(i)}$;   Update the features with false attribution.

19:       **end for**

20:    **end if**

21:    **for** $((\tilde{x}^{(i)}, y^{(i)}, \mathcal{A}_{\text{inc}}^{(i)}, \Delta\mathcal{A}^{(i)}), \hat{\mathcal{A}}^{(i)})$ in $(\tilde{\mathbb{D}}_m, \{\hat{\mathcal{A}}^{(i)}\}_{i=1}^{N})$ **do**

22:       $q^{(i)} \leftarrow \frac{|\mathcal{A}_{\text{inc}}^{(i)}| - |\hat{\mathcal{A}}^{(i)}|}{|\mathcal{A}_{\text{inc}}^{(i)}|}$;

23:    **end for**

24:    $\bar{q} \leftarrow \frac{1}{N} \sum_{i=1}^{N} q^{(i)}$   // Attribution ratio at the current step.

25:    $\text{Append}(\mathbb{P}, (s_m, \bar{q}))$   // Update results.

26:    $s_0 \leftarrow s_m$;   // Update the accuracy at the previous step.

27:    $m_0 \leftarrow m$;   // Update the mask ratio at the previous step.

28: **end for**

   **Return**: $\mathbb{P}$

---

assigned (as described in the main text). The model is a linear model and can be formulated as $y = \sigma(\sum_i x_i)$, where $x_i$ is the $i$-th feature, and $\sigma(\cdot)$ is a step function that rises at $x = 0$, $\sigma(x) = -1$ if $x < 0$, and $\sigma(x) = 1$ if $x > 0$. In other words, the model also sums up all features of the input and returns a positive value if the result is greater than zero. Hence, the model can classify the dataset with 100% accuracy. Lastly, we describe how we create ground-truth attribution maps for this model and dataset. As the model is a linear model, and each feature $x_i$ is sampled from a zero-mean Gaussian distribution, the Shapley value for $x_i$ with $\sigma(x_i) = 1$ is then $1 \cdot (x_i - \mathbb{E}[x]) = x_i$. Similarly, the Shapley value for $x_i$ with $\sigma(x_i) = -1$ is $-x_i$. We confirm that attribution maps generated by Shapley values are fully correct for linear models. As a result, for positive samples, the attribution values are the same as feature values. For negative samples, the attribution values are the negation of feature values, which means that negative features actually contribute to the negative decision. Finally, we modify the attribution maps to be compatible with our soundness and completeness evaluation. Since our evaluation only supports positive attributions, we clip negative attribution values to zero. This conversion step has no negative effect on the actual evaluation. The rest of the evaluation setup is identical to other experiments.

---

**Algorithm 4** Completeness Evaluation

---

1: **Input:** $f$: model; $\mathbb{D} = \{x^{(i)}, y^{(i)}\}_{i=1}^N$: labeled dataset and attribution maps $\{\mathcal{A}^{(i)}\}_{i=1}^N$ for each $(x_i, y_i)$; $\phi$: perturbation function; $\psi$: noisy linear imputation function; $\mathbb{T} = \{0.9, 0.8, 0.7, \ldots, 0.1\}$: attribution thresholds; Accuracy: accuracy evaluation function.
2: **Output:** $\mathbb{S}_\Delta$: Accuracy differences associated with attribution thresholds $\mathbb{T}$.
3: Initialize $\tilde{\mathbb{S}}_\Delta \leftarrow [\,]$;
4: $s_0 \leftarrow \text{Accuracy}(f, \mathbb{D})$;   // Accuracy on the unperturbed dataset
5: **for** $t$ in $\mathbb{T}$ **do**
6:    $\tilde{\mathbb{D}}_t \leftarrow [\,]$;   // Initialize dataset of imputed images
7:    **for** $(x^{(i)}, y^{(i)}), \mathcal{A}^{(i)}$ in $(\mathbb{D}, \{\mathcal{A}^{(i)}\}_{i=1}^N)$ **do**
8:       $\hat{x}^{(i)} \leftarrow \phi(x^{(i)}, \mathcal{A}^{(i)}, t)$;   // Perturb pixels whose attribution exceed $t$
9:       $\tilde{x}^{(i)} \leftarrow \psi(\hat{x}^{(i)})$;   // Impute the perturbed image
10:       $\text{Append}(\tilde{\mathbb{D}}_t, (\tilde{x}^{(i)}, y^{(i)}))$;
11:    **end for**
12:    $s_t \leftarrow \text{Accuracy}(f, \tilde{\mathbb{D}}_t)$;   // Accuracy on the imputed dataset
13:    $\text{Append}(\mathbb{S}_\Delta, s_o - s_t)$;   // Accuracy difference at the current step
14: **end for**
15: **Return** $\mathbb{S}_\Delta$

---

| Implementation | Removing attribution | Introducing attribution |
|---|---|---|
| Constant | Subtract the attribution map by a constant 0.6. | Add the attribution map with a constant 0.6. |
| Random | Sample a shift from $\mathcal{U}(-0.6, 0)$ for each pixel independently, and add the attribution of each pixel with the corresponding shift. | Sample a shift from $\mathcal{U}(0, 0.6)$ for each pixel independently, and add the attribution of each pixel with the corresponding shift. |
| Partial | Sort the attribution map in ascending order and select the pixels in the indexing range $[0.6N, 0.8N]$, where $N$ is the number of pixels. Then set the attribution of these pixels to 0. | Sort the attribution map in ascending order and select the pixels in indexing range $[0, 0.4N]$, where $N$ is the number of pixels. Then set the attribution of these pixels to $q_{0.8}$, where $q_t$ denotes the $t$-th quantile of attribution values. |

Table 3: The modifications of attribution maps on ImageNet.

## G.2 ImageNet images for feature attribution

We randomly select 5 images for each class in the ImageNet validation set, obtaining a subset with 5000 images. When performing attribution, the images and attribution maps are resized to $224 \times 224$ before being fed into the pretrained VGG16 model.

This subsection presents the configurations for generating attribution maps on ImageNet. For GraCAM, We resize the resulting attribution maps to the same size as the corresponding input images. We use the implementations of GradCAM, DeepSHAP, IG, IG ensembles in Captum (Kokhlikyan et al., 2020). Some hyper-parameters for producing attribution maps are:

- **GradCAM** We perform attribution on the `features.28` layer of VGG16 (i.e. the last convolutional layer).

- **DeepSHAP, IG and IG ensembles** We choose 0 as the baseline for attribution. We clamp the attribution to $[0, 1]$.

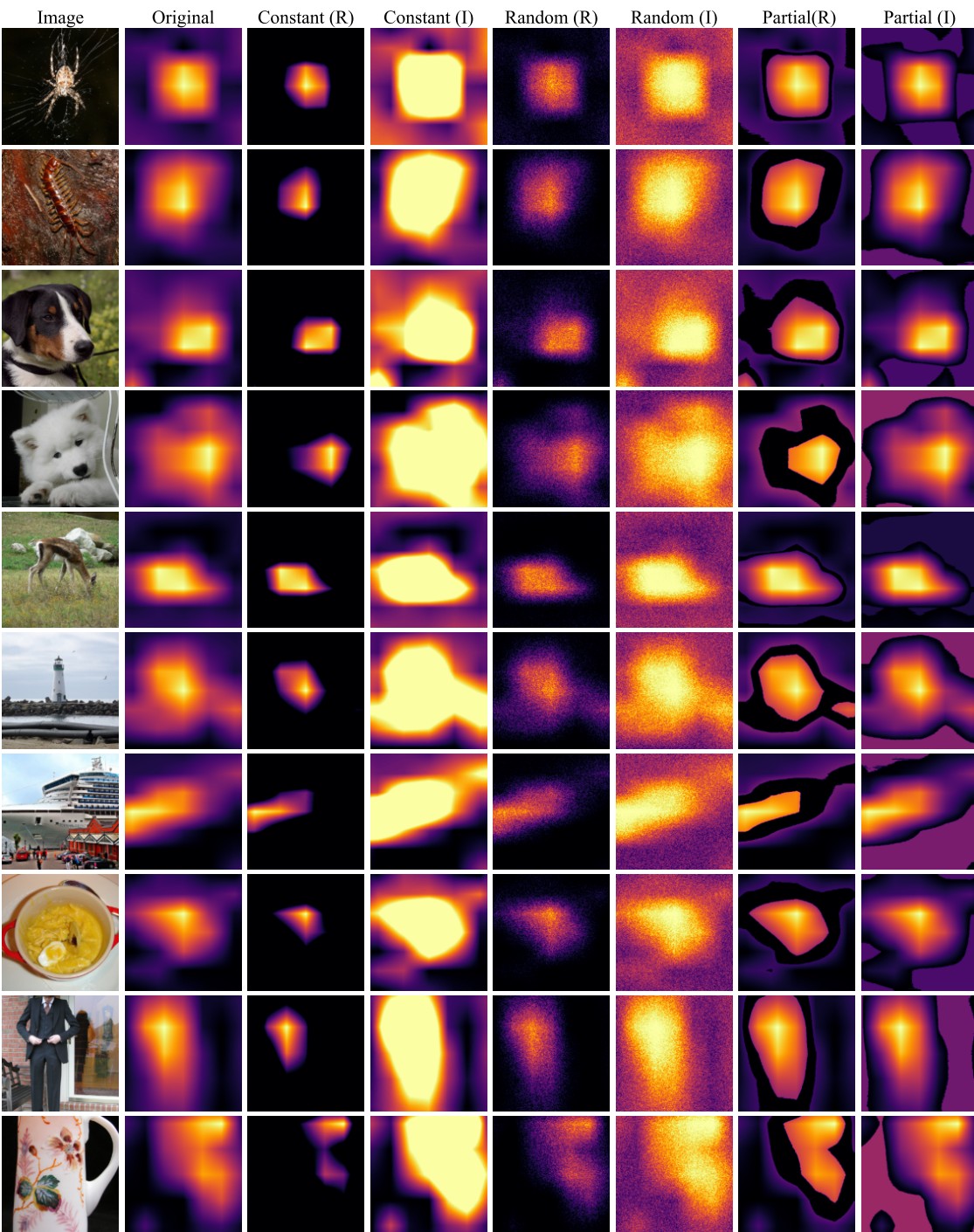

Figure 13: Examples of modified attribution maps. (R) and (I) denote the *Remove* and *Introduce* modification, respectively. The authors ensure that samples are not cherry-picked.

## G.3 Modifications of attribution maps on ImageNet

We implement three different modification schemes and modify the attribution maps. The details thereof are presented in Table 3. We show some examples of modified attribution maps in Figure 13. The images are randomly selected.

| Image | IG | IG-SG | IG-SQ | IG-Var |
|-------|-----|--------|--------|--------|

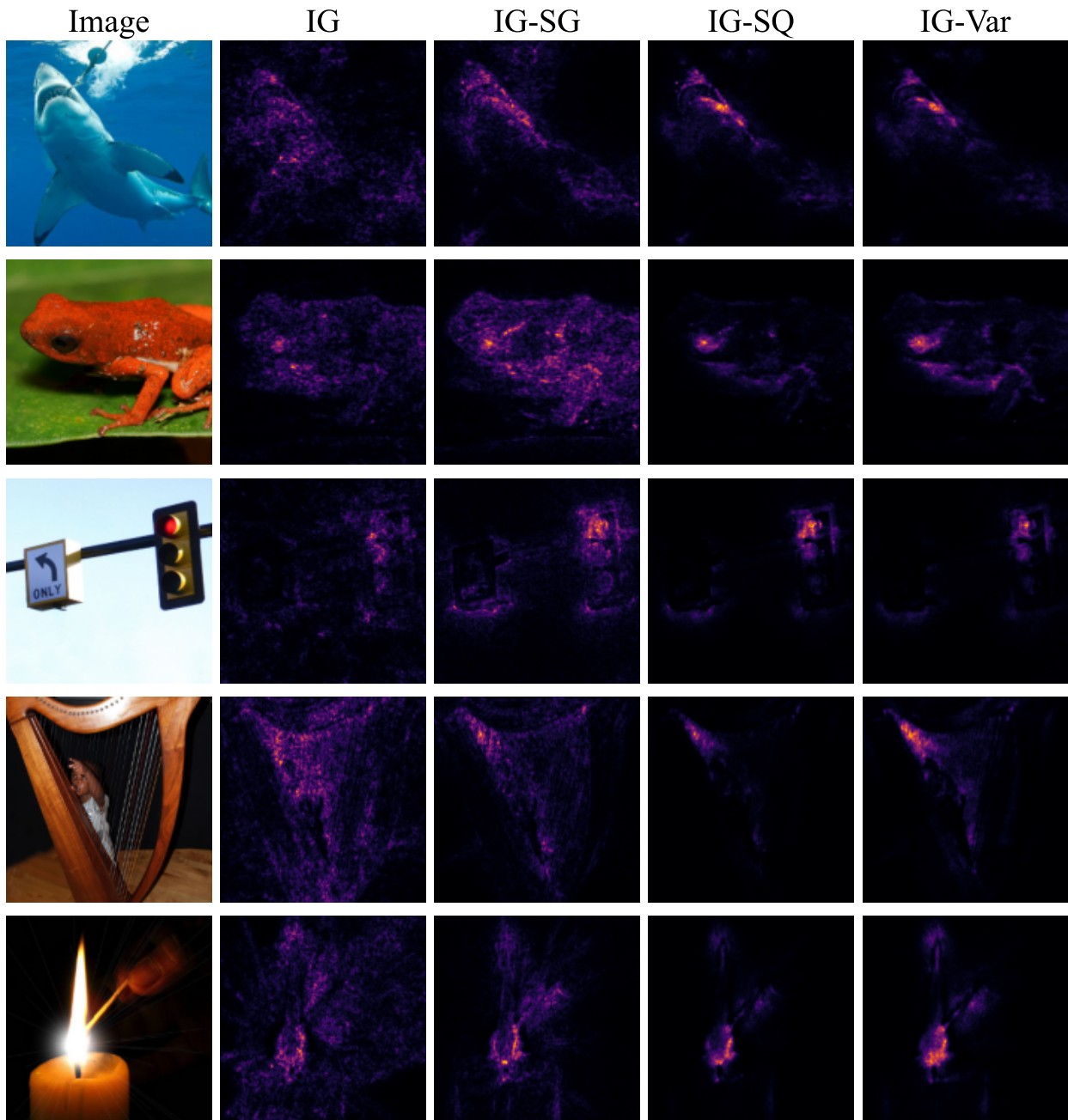

Figure 14: Examples of IG and IG ensembles attribution maps. The authors ensure that samples are not cherry-picked.

## G.4  Benchmarking Ensemble Methods of IG

In each ensemble method, we use an isotropic Gaussian kernel $\mathcal{N}(\mathbf{0}, 0.3 \cdot \mathbf{I})$ to sample 20 noisy samples for a given input sample. Figure 14 shows additional examples of IG and its ensemble methods. The images are randomly selected.

In addition, we also compare IG and IG ensembles using ROAD in LeRF order, and the result is shown in Figure 15. For convenience, we copy the figures of other metrics from Figure 8 and paste them in Figure 15.

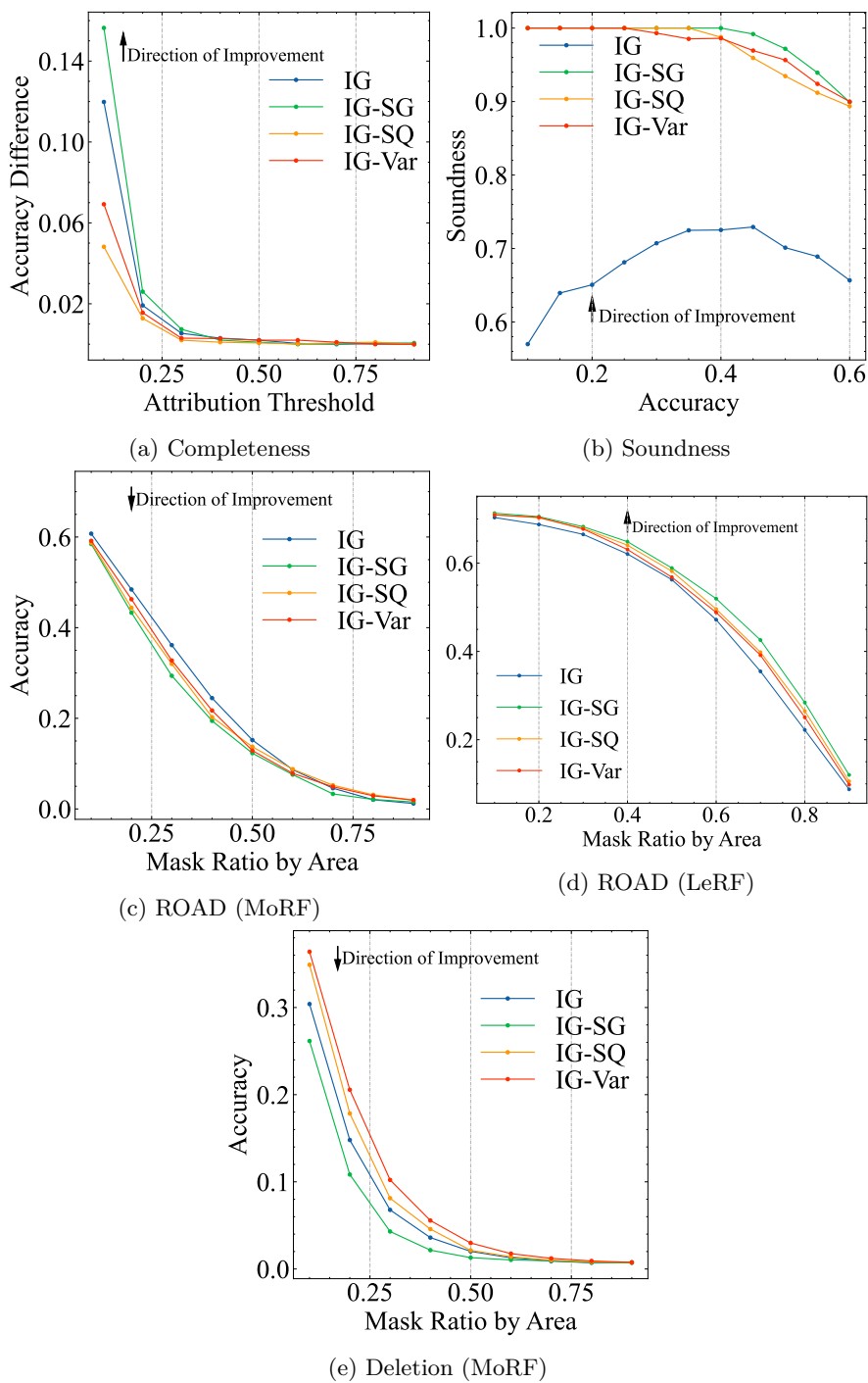

Figure 15: Evaluations of IG ensembles. Not all ensembles improve the completeness (a) of IG, but they significantly improve soundness (b). However, the advantage of ensemble methods over IG is not notable in ROAD (c)-(d) or Deletion (e) compared to that in soundness.

Similar to ROAD in MoRF order, IG ensembles do not show a considerable advantage in the results of ROAD in LeRF order.

