# OpenReview forum: "A Dual-Perspective Approach to Evaluating Feature Attribution Methods"
_TMLR — Accepted by TMLR_

### Review · Reviewer_CkYd · 2024-04-30

**Summary Of Contributions:**

The paper "A Dual-Perspective Approach to Evaluating Feature Attribution Methods" proposes a new method for evaluating feature attribution methods. It proposes to decompose the evaluation in two distinct metrics, namely soundness, which refers to precision where we look at the ratio of attributed features that is included in the true predictive feature set, and completness, refering to recall which assess the rate of true predictive features that are attributed by the method. It looks at better ranking attribution methods than existing methods from the litterature.

**Audience:**

Yes

**Broader Impact Concerns:**

.

**Claims And Evidence:**

No

**Requested Changes:**

As said in previous section, I which authors could further detail competitor methods (such as ROAD, ROAR and deletion), with som minimal formalization of their principle to help the reader to better understand differences. Also, the global objective of the task should be formalized clearly at the begining of the paper. The procedure "to mitigate OOD effects" should also be detailled in the main paper, as it is core of the proposed method.

Also, the differences with classical MoRF and LeRF should be further hilighted.

One important concern I have lies in the approximation method for finding soundness, which is based on an iterative algorithm (that could be clearer) that sequentially introduces features in a set until reaching a threshold accuracy value. Such an algorithm is clearly not guaranteed to find the set A* targeted by the method. Sequentially discarding features that did not contribute enough together with already selected features (less than epsilon, which is not introduced and discussed anywhere in the text) clearly do not guarantee that they would have not contributed more with other  selected components. It assumes sub-modularity of the accuracy, which clearly does not look as a realistic assumption. Please discuss.


Minors :

 - At the bottom of page 6, I did not understand sentence "Specifically, the model...  ", which look counter-intuitive to me
 - I did not well understand why authors state in the proof of theorem 4.11 that S\interI=A_inc \inter I, as S is a subset of A_inc...
 - Evaluation with semi-natural datasets consider a caricatural example to illustrate inconsistency. But I suppose that it can exist some clever methods to introduce ground truth features in the input, which would be more consistent. Also, the motivating experiment with retrain is not fully convincing. I wish we could get a working example together with such setting that induces spurious correlations

**Strengths And Weaknesses:**

Strengths:
  - Important problem, which is particularly challenging
  - Decomposition in precision/recall metrics looks relevant
  - Rather simple method to implement

Weaknesses:
   - Some parts are the paper are not sufficiently detailled or not enough from my point of view. Competitor methods should be further detailed to help the reader with a self-contained paper. Motivating experiments could be more clear.
    - Implemented methods are not fully in line with motivations, as considering accuracy as a proxy for assessing a set of features (assumption 4.10) can induce misleading rankings. Also, approximation made to assess the soundness metric look really not tight. I am not fully confident with proposed methods.
    - I cannot fully capture the difference between the proposed completeness and the LeRF method

---

> ### Author Response · Authors · 2024-07-27
> **Response to Reviewer CkYd**
>
> Thank you for acknowledging that our paper addresses an important problem in feature attribution and the relevance of decomposing evaluation into precision/recall metrics. We have addressed your concerns as follows:
>
> **Additional details**: Thank you for the suggestion. Now we have included more details about competitor methods and added further motivation for our method, as suggested by you and the Reviewer 67M7. Please see Section 2 as well as the first paragraph of Section 4 in the revised manuscript. The core of this work is to reveal some limitations of previous evaluation metrics (order-based evaluation, synthetic data, retraining) and to propose a metric that evaluates attribution maps similar to a precision/recall manner to provide insights on improving attribution methods. Thus, we do not consider mitigating OOD effects as our contribution.
>
> **Difference compared to MoRF and LeRF**: (1) The soundness evaluation differs fundamentally from classical Insertion/Deletion metrics. While soundness evaluates the ratio of attribution values associated with two feature sets, Insertion/Deletion assesses accuracy following feature removal and employs attribution values solely for feature sorting. (2) The completeness evaluation diverges from the Deletion/Insertion approach in its method of feature removal: it removes features with attribution exceeding a specific value, whereas Insertion/Deletion removes features whose ranking is better than a certain threshold. We have incorporated this discussion in the revised manuscript. Please refer to the last paragraphs in the revised Section 4.2 and Section 4.3.
>
> **Approximation method for finding soundness**: Thank you for highlighting this important aspect of our work. Our main proposal is defining soundness and a theoretically correct evaluation framework with the flexibility to apply any performance metric. For implementation, we chose accuracy as the performance metric due to its widespread use in other evaluation metrics like ROAD and ROAR. We acknowledge the limitation you mentioned that a feature might contribute more to another feature set. To best mitigate this issue, we employed two strategies: (1) including features from highest to lowest attribution and stopping at a certain accuracy threshold before saturation to include potentially impactful features, and (2) including batches of features instead of one at a time to compare significant accuracy changes. We have added more discussion on this in the last paragraph in Section 4.2. We want to emphasize that we do not restrict the choice of performance metric in our theoretical framework.
>
> **Minors**:
>
> * Regarding Assumption 4.10, we have clarified that while model performance is suitable for comparing feature sets to determine which contains more predictive information, it is not appropriate for measuring the precise difference in information. We have added a sentence to explicitly address this point. Please refer to the last paragraph of Section 4.1 for further details.
>
> * We conclude this based on the condition \rho(f(S)) = \rho(f(Ainc)) in theorem 4.11.
>
> * Experiments in Section 3 aim to falsify prior works and can effectively highlight their issues. A systematic and comprehensive study of how these evaluation metrics behave could warrant a separate paper and will consider this for future research.
>
> In response to requested changes:
>
> * We address the concerns in the points above.
>
> * The manuscript has been revised correspondingly as described in the points above.

---

> > ### Comment · Reviewer_CkYd · 2024-07-29
> > **.**
> >
> > > "The completeness evaluation diverges from the Deletion/Insertion approach in its method of feature removal: it removes features with attribution exceeding a specific value, whereas Insertion/Deletion removes features whose ranking is better than a certain threshold."
> >
> > ok thanks but why would it be better ? do you have some theoretical justification, or at least informal intuition ?

---

> > ### Comment · Reviewer_CkYd · 2024-07-29
> > **.**
> >
> > > "including batches of features instead of one at a time to compare significant accuracy changes"
> >
> > how these batches are sampled ? wouldn't it be possible to average results over multiples runs of soudness computation, with random sampling of batches ?
> >
> > Is it possible to design an assesment of the bias induced by the proposed approximation ?
> >
> >  > "including features from highest to lowest attribution and stopping at a certain accuracy threshold before saturation to include potentially impactful features"
> >
> > In that setting, isn't it very similar to addition methods that include features from most to least attributed ones ?

---

> > > ### Author Response · Authors · 2024-07-30
> > > **2. Response to Reviewer CkYd**
> > >
> > > >how these batches are sampled ? wouldn't it be possible to average results over multiples runs of soudness computation, with random sampling of batches ?
> > >
> > > Features are sampled in batches based on the descending order of their attribution values. The efficacy of using random sampling and Monte-Carlo simulation to mitigate bias remains uncertain. Given the large number of features, a correspondingly high volume of sampling trials would be necessary to encompass all potential cases. However, conducting a single trial for soundness evaluation takes approximately 48 hours, primarily due to time consumption of the imputation operation. Thus, from a practical standpoint, our current strategy is more viable.
> > >
> > > Additionally, designing an effective method to quantify the approximation bias presents its own set of challenges. This is particularly true as quantifying feature interaction is still an unresolved and complex issue within the field of explainability.
> > >
> > > >In that setting, isn't it very similar to addition methods that include features from most to least attributed ones ?
> > >
> > > The soundness metric fundamentally differs from the Insertion metric in its implementation and theoretical basis. We progressively expand the feature set from highest to lowest attribution values, while simultaneously monitoring an informative subset of features and calculating **the ratio of attributions** between this subset and the expanded set. In contrast, the Insertion metric simply involves adding features and measuring **the resulting accuracies**, making the evaluation processes and computed targets quite distinct. Details of the soundness evaluation process are delineated in Algorithm 1. Although the order of feature inclusion is similar between Soundness and Insertion in our implementation, our theoretical framework does not confine us to this choice of feature inclusion. As previously mentioned, one can also explore alternative methods for expanding the feature set.

---

> > > > ### Comment · Reviewer_CkYd · 2024-07-31
> > > >
> > > > Ok thanks I better understand now
> > > >
> > > > But what takes 48 hours as you mentionned in your answer ?
> > > >
> > > > I suggested to perform monte carlo estimations, by sampling and averaging, not being exhaustive in the exponentially huge set of combinations... from my point of view using sampling strategies would give more robust estimations than simply taking by descending order of attribution values.

---

> > > > > ### Author Response · Authors · 2024-07-31
> > > > > **.**
> > > > >
> > > > > A single run for Soundness evaluation currently takes approximately 48 hours, primarily due to the time required for feature imputation. The feature imputation requires solving large sparse linear equations. However, we anticipate significant reductions in imputation time when NVIDIA implements cuSOLVER for solving sparse equations using CUDA. With such hardware acceleration or other imputation methods, employing Monte-Carlo simulations and random sampling could become a viable option. We appreciate the reviewer’s suggestion to explore Monte-Carlo estimation to ehance our method.

---

> > > > > > ### Comment · Reviewer_CkYd · 2024-08-01
> > > > > >
> > > > > > Ok... I thus realize that imputation is an very important part of your approach, while you only mention it very succintly in the main paper (and only mentionned in algorithms in the appendix). Even its justification / intuition abvout why this is needed is not really discussed. Do you have abblation experiments without this imputation ? This looks very important as it is core of your approach...
> > > > > >
> > > > > > Also, this looks to be a strong limitation of the approach, as it appears to not scale well. The imputation complexity depends on the number of features ?

---

> > > > > > > ### Author Response · Authors · 2024-08-01
> > > > > > >
> > > > > > > Imputation is a well-known technique proposed by ROAD [1] to mitigate class information leakage through masking. As analyzed in [1], any feature attribution evaluation involving masking or perturbation can introduce bias due to class information leakage. Therefore, the authors of [1] suggest performing imputation after masking. While our research question is unrelated to solving class information leakage and does not overlap with [1], we acknowledge the importance of prior work in this area. Consequently, we have included imputation in our methodology since we perform masking during evaluation.
> > > > > > >
> > > > > > > We discuss [1] in our Related Works section and clearly state at the end of Section 4.2 that we perform imputation as recommended by the authors of [1]. We do not think an ablation study on imputation is necessary, as this experiment would only demonstrate the impact of class information leakage, which is already thoroughly discussed in [1]. Additionally, the time consumption of imputation is a limitation of [1] rather than our method. If future research proposes better ways to mitigate class information leakage, our method can readily integrate those improvements.
> > > > > > >
> > > > > > > [1]: Rong, Y., Leemann, T., Borisov, V., Kasneci, G., & Kasneci, E. (2022). A consistent and efficient evaluation strategy for attribution methods. ICML 2022.

---

> ### Author Response · Authors · 2024-07-30
> **2. Response to Reviewer CkYd**
>
> >ok thanks but why would it be better ? do you have some theoretical justification, or at least informal intuition ?
>
> Removing features based on attribution values allows the completeness metric to capture not only differences in attribution rankings but also variations in attribution values when comparing two attribution maps. If two attribution maps rank features identically but assign significantly different attribution values, conventional metrics may fail to distinguish between them. However, our completeness metric effectively resolves this issue. We validate this assertion with empirical results presented in Figure 7 and Table 1. Additionally, we illustrate this concept with a straightforward example in Section 3.3 of the revised manuscript.

---

### Review · Reviewer_67M7 · 2024-06-03

**Summary Of Contributions:**

The paper propose two evaluation metrics (soundness and completeness) for feature attribution methods that quantify the contribution of input features to a model's output.
Soundness measures the degree to which attribution methods assign importance to "predictive" features, whereas completeness quantifies the extent to which the attribution method collects and assign values to "predictive" features.
The authors provide algorithms to compute these metrics and clarify required assumptions.

**Audience:**

Yes

**Claims And Evidence:**

Yes

**Requested Changes:**

* Please address the weaknesses points above
* It is unclear what Expand in line 4 does.
* There are many citations that should be parenthetic (e.g., the very first sentence). These should be corrected.
* What is the attribution method used for Figure 7.(b)-(e)?
* Appendix G.1:  Is the Shapley value the attribution method?

**Strengths And Weaknesses:**

## Strengths
* The authors reveal limitations with existing methods that are based on retraining with removed attributes and on injected ground-truth features.
* The proposed metrics can capture differences in terms of the assignment of attribute importance, which existing methods that depend on the importance ranking fail to detect.
* The proposed method does not require retraining of models and creating additional datasets.

## Weaknesses
* It is not clear how the limitations of the existing work have been addressed by the proposed metrics (the connection between Section 3 and 4 are not so clear).

* The paper is not as self-contained as it could be:
   1. Evaluation strategies like ROAD and Deletion are not explained at all.
   2. The same applies to feature attribution methods used in the experiments (such as IG) -- the ROAR paper does a good job in this regard.

Minor comments:
* The definition of "predictive information" is ambiguous. "Predictive" can be interpreted as being useful to achieve high performance in some prediction task or features that are used by a model to make certain decisions (but are not necessarily helpful in a given task). It appears that Assumption 4.10 effectively defines what informative means in terms of a performance metric $\rho$.
* Definition 4.2 does not seem to define what a attribution method is (is just a function on $\mathcal{F}$ that outputs nongevative values?).
There is also a typo: "a feature $F$ and a feature $F$".

---

> ### Author Response · Authors · 2024-07-27
> **Response to Reviewer 67M7**
>
> We thank the reviewer for acknowledging our paper’s contribution in revealing the limitations of previous evaluation methods and that our method can capture differences in attribution while other methods cannot. We address the concerns as follows:
>
> * W1: As mentioned in the beginning of Section 4, our designed method does not require retraining a model or creating a modified dataset. Hence, our evaluation method does not have issues identified in the Section 3 due to retraining and semi-natural datasets. Nevertheless, we just mentioned it in one sentence. We value your feedback and have expanded this sentence to provide a more comprehensive motivation on the design of our method. Please see the first paragraph in Section 4.
>
> * W2: Thanks for your valuable suggestion. We revised our text and included more description on other evaluation metrics and feature attribution methods. Please refer to Section 2.1 and Section 2.2.
>
> **Minor**:
>
> * M1: In the context of supervised learning, where a fixed task and ground truth labels are provided, “predictive information” refers to the information used by the model to make predictions that closely approximate these labels. For example, in image classification, if an image is labeled as “dog,” the predictive information consists of the visual details used by the model to classify the image as “dog”. This information is directly linked to the ground truth labels and indirectly related to performance metrics. In scenarios like image classification, predictive information is defined by the class labels. Metrics such as accuracy, Brier score, and F1 score evaluate how well the model’s predictions align with these labels, so richer predictive information leads to higher performance scores. In summary, the ambiguity is addressed by directly associating predictive information with ground truth labels.
>
> * M2: An attribution method can be seen as a function that assigns an attribution value to a feature within a fixed model. We appreciate the reviewer’s attention for pointing this out. We have revised Definition 4.2.
>
> In response to requested changes:
>
> * We addressed the concerns in the points above.
>
> * Expand in Line 4 is to include more features into the set $A_inc$ (inc stands for “include”). We gradually expand $A_inc$ and then approximately identify the subset of features that are truly predictive, and then we compute the ratio between attribution values of this predictive subset and the attribution values of the included features $A_inc$.
>
> * We have fixed the citation issue in the revised manuscript.
> * For Figure 7, GradCAM is used as a baseline attribution method.
> * Yes, Shapley value is the attribution method in G.1.
>
>
> [1] Xue, A., Alur, R., & Wong, E. (2023). Stability guarantees for feature attributions with multiplicative smoothing. Advances in Neural Information Processing Systems, 36.

---

### Review · Reviewer_RoJR · 2024-07-13

**Summary Of Contributions:**

See 'official comment' already filed.

**Audience:**

Yes

**Broader Impact Concerns:**

See 'official comment' already filed.

**Claims And Evidence:**

Yes

**Requested Changes:**

See 'official comment' already filed.

**Strengths And Weaknesses:**

See 'official comment' already filed.

---

### Comment · Reviewer_RoJR · 2024-07-11

**Summary of contributions**

Given a feature set $\mathcal{F}$, let $\mathcal{I}$ be the subset of features containing predictive information, and $\mathcal{A}$ the subset with positive attributions under some feature attribution technique.

The authors introduce two new measures for feature attribution, _soundness_ and _completeness_: $\mathcal{A}$ is _sound_ if $\mathcal{A} \subseteq \mathcal{I}$, _complete_ if $\mathcal{I} \subseteq \mathcal{A}$ and _optimal_ if $\mathcal{A} = \mathcal{I}$.

The authors demonstrate these measures first on a synthetic dataset with IID random features, and a target derived linearly from them.  Specifically, they conduct two experiments:
- _Introduce_: increases the attribution values of non-predictive features, lowering soundness while improving completeness; and
- _Remove_: removes the attribution values [of which?], lowering completeness without influencing soundness.

_Remove_ outperforms ground-truth attribution in the sense that, when features above a given attribution level are removed, it displays smaller loss in accuracy relative to the unperturbed model.  In turn, ground-truth attribution outperforms _Introduce_.

For soundness, ground-truth attribution and _Remove_ maintain soundness as model accuracy is reduced; _Introduce_ does not, only climbing back towards 1.0 for very low levels of accuracy.  Thus, "the evaluations behave as expected".

On non-synthetic data, _Remove_ and _Introduce_ are again applied, this time to attribution maps generated by GradCAM.  The authors claim that a measure is effective if the curves of the evaluation results for these methods (and ground-truth) are non-overlapping.

The authors also demonstrate that the difference between Integrated Gradients and ensembles based on IG almost entirely reflects differences in soundness.  ROAD and Deletion do not show clear differences.

This is found for Completeness and Soundness, but not for ROAD or Deletion.

**Strengths and weaknesses**

_Strengths_
- understanding feature attribution is vital, particularly as models become larger and more complex
- to me, this analysis seems novel
- further, I do not see flaws in the authors' arguments

_Weaknesses_
- I don't have strong intuitions around feature attribution in image models, so need clearer arguments to convince me: for me, a stronger paper would have cut some of the preamble, and focused on explaining the main arguments more slowly, carefully and clearly.
- specifically: the _Introduce_ and _Remove_ techniques seem designed particularly to align with _Soundness_ and _Completeness_.  Thus, the force of using these for validation is unclear.  I would have liked to see a proper consideration of this: presumably ROAD and Deletion were validated against tests of their own.  Which were these?  What happens if they are also included in the analysis?
- relatedly, the use of IID rvs in the synthetic example rules out interesting 'Rashomon set' questions - in which multiple features can be used equally well.
- overall, the approach felt underspecified: an attributed feature set is `optimal' if it corresponds to the predictive feature set - with no further restriction on the strengths.  I found myself wondering how these axioms compared to e.g. Shapley axioms - which identify a unique measure.
- given my lack of strong intuitions in general, I would find helpful an example showing a mistake that e.g. a business risk manager would understand that is made by other approaches, but not the present one.
- "Since our evaluation only supports positive attributions" (p.20): this seems an important limitation.
- minor: \citep{} should be used in some of the places where \cite{} or \citet{} currently is.
- minor: I found the use of $f, F$ and $\mathcal{F}$ very non-intuitive

**Requested changes**
I would like to see improvements on the areas of weakness above, other than those identified as 'minor'.

**Broader impact concerns**
None.

---

> ### Author Response · Authors · 2024-07-27
> **Response to Reviewer RoJR**
>
> Thank you for acknowledging the contributions of our paper and providing constructive comments. We address the raised points as follows:
>
> * W1: We have expanded the related work section and included a more comprehensive background on feature attribution methods applicable to vision models. (Please see Section 2.1 and 2.2.)
> * W2: The tests tailored for ROAD etc. are not applicable to our metrics. Therefore we design our own tests for our metrics. The details are as follows:
>
>     * **What tests are used by existing metrics**: The existing metrics e.g. ROAD or ROAR typically assess the consistency of the evaluated attribution methods’ rankings under two different feature removal orders: Most Important First (MoRF) and Least Important First (LeRF).
>
>     * **Why we design our own test instead of using theirs**: The test introduced by ROAD etc. is unsuitable for our metrics because: (1) Soundness and Completeness evaluate different characteristics of an attribution method. For example, a method that is highly sound may exhibit suboptimal completeness, making it inappropriate to harness previous consistency tests to validate our metrics. (2) ROAD etc. focus solely on the correct order of features—whether an attribution method accurately determines that feature A is more salient than feature B. Our metrics, in contrast, consider the attribution values. This point is also acknowledged by Reviewer 67M7 in the Strength 2 in his/her comments. To validate that our metrics can effectively capture variations in attribution values, we need more fine-grained tests. To this end, we proposed Introduce and Remove, which modifies the attribution values while maintaining the order of features.
>
> * W3:  In the synthetic example,  we manually design both the model’s weights and the data to obtain ground truth attribution for testing our metrics. The Rashomon effect can pose a challenge if models are **trained** on this synthetic data; such trained models could use different subsets of features to achieve the same performance level. This variability could lead to uncontrollable attribution values, obscuring which features are truly influencing the model’s decisions. In contrast, our designed model ensures that attribution values are fixed and known, once the model weights and data are established.
>
> * W4: Definition 4.6 delineates the optimal case where sets $\mathcal{A}$ and $\mathcal{I}$ contain identical features (or equivalently, identical feature indices). This case is uniquely characterized by the matching of feature indices. However, considering only the indices of features is insufficient for fully assessing feature attribution methods. Two attribution methods may identify the same attributed feature set yet assign different attribution values to these features. To address this, we introduce the properties of soundness and completeness in Definitions 4.8 and 4.9, respectively. These properties are sensitive to the attribution values, akin to the considerations in the Shapley axioms. Now, we have expanded the motivation for our proposed metrics in the paragraph following Definition 4.6. Please refer to  the revised manuscript for further details.
>
> * W5: Thanks for your valuable suggestion. In the revised manuscript, we have added Section 3.3 to show a failure case of other metrics, and the shown example is in tabular data domain rather than in computer vision. The example reveals the fundamental limitations of order-based evaluation metrics.
>
> * W6: Many attribution methods, such as GradCAM, Extremal Perturbation, IBA, and InputIBA, primarily identify features that contribute positively, whereas others, including IG and DeepSHAP, are capable of recognizing both positive and negative contributions. To facilitate a fair comparison across all these methods, we have designed our metrics to initially evaluate only the positively attributed features. However, our metrics are adaptable and can be extended to include negatively attributed features.
> Minor points: We thank the reviewer for the suggestions. In response, we have revised the manuscript to incorporate the suggestions.
>
> In response to requested changes:
>
> * We address the concerns in the points above.
> * The manuscript has been revised correspondingly.

---

### Author Response · Authors · 2024-07-27
**Looking Forward to Discussion**

Dear reviewers,

We have posted our responses and uploaded a revised version of our manuscript. We would like to invite you to join discussions in case you still have concerns and questions. Look forward to interactive discussions.


Best,

Authors

---

### Decision · Action_Editor_TvUw · 2024-10-31

**Recommendation:** Accept with minor revision

**Comment:**

I think this works is interesting and (especially) useful. While I agree with some of the reviewers' comments that the method doesn't fully disambiguate the confounding effects (weakness of evaluation method vs. weakness of the underlying model), the bar that is set there is too high IMO. I view this work not as a self-contained method that gives us all the answers, but as simply another tool that a researcher or practitioner can use to understand and verify their hypotheses.

I think the work can be accepted and the authors should heed the reviewers' comments on clarifying the underlying methods more (specifically, would be good to address Reviewer RoJR's comments about Sections 3 & 4).

**Audience:**

Anyone interested in attribution methods and how they compare, which is a larger audience of TMLR.

**Claims And Evidence:**

The work proposes two metrics (soundness and completeness) for analyzing/understanding feature attribution methods. Soundness is akin to precision and completeness is akin to recall. The idea is to use these metrics to rank attribution methods in the literature.

In general, this is an interesting and important problem, as it's not obvious how to actually understand the empirical pros and cons of various attribution methods. The method itself appears easy to implement, sound from a methodological POV and novel. Importantly, the method doesn't require the user to retrain models of create new datasets.

---

> ### Author Response · Authors · 2024-11-24
> **Thank you for your feedback and guidance, and we have uploaded the camera-ready version.**
>
> Dear AE and Reviewers,
>
>
>
> We deeply appreciate your valuable feedback and constructive comments, which have significantly enhanced the quality of our work. Based on your insightful suggestions, we have made several improvements to the camera-ready manuscript. Below, we summarize the key updates:
>
> * **Section 2:** We have expanded the *Related Work* section to include more detailed discussions on existing feature attribution methods and the metrics used to evaluate them.
>
> * **Section 3:** A new subsection, *Section 3.3*, has been added to highlight the limitations of order-based evaluation metrics through a simple synthetic example. This provides readers with a clearer intuition for our proposed evaluation approach.
>
> * **Section 4:** At the beginning of the section, we have elaborated on the motivation derived from previous approaches and clarified the core intuition behind our method.
>
> * **Section 4.1:** We have included additional explanations about the definition of the optimal attributed feature set, further elaborating our motivation and enhancing the rigor of our approach.
>
> * **Section 4.2:** We have expanded the discussion on how *Algorithm 1* approximates actual soundness, providing greater depth and clarity.
>
> Thank you once again for your thoughtful guidance and support during the review process. We hope that the revisions have enhanced the clarity of the manuscript. If there are further suggestions to improve the manuscript, we would be grateful to hear your ideas.
>
> Best regards,
>
> Authors of the paper